EMBO
Molecular Medicine

# FBXL4 deficiency increases mitochondrial removal by autophagy

David Alsina[1,2,†], Oleksandr Lytovchenko[1,2,†], Aleksandra Schab[1], Ilian Atanassov[3], Florian A Schober[2,4], Min Jiang[5], Camilla Koolmeister[1,2], Anna Wedell[4,6], Robert W Taylor[7], Anna Wredenberg[1,2,6] & Nils-Göran Larsson[1,2,6,*]

## Abstract

Pathogenic variants in *FBXL4* cause a severe encephalopathic syndrome associated with mtDNA depletion and deficient oxidative phosphorylation. To gain further insight into the enigmatic pathophysiology caused by FBXL4 deficiency, we generated homozygous *Fbxl4* knockout mice and found that they display a predominant perinatal lethality. Surprisingly, the few surviving animals are apparently normal until the age of 8–12 months when they gradually develop signs of mitochondrial dysfunction and weight loss. One-year-old *Fbxl4* knockouts show a global reduction in a variety of mitochondrial proteins and mtDNA depletion, whereas lysosomal proteins are upregulated. Fibroblasts from patients with FBXL4 deficiency and human *FBXL4* knockout cells also have reduced steady-state levels of mitochondrial proteins that can be attributed to increased mitochondrial turnover. Inhibition of lysosomal function in these cells reverses the mitochondrial phenotype, whereas proteasomal inhibition has no effect. Taken together, the results we present here show that FBXL4 prevents mitochondrial removal via autophagy and that loss of FBXL4 leads to decreased mitochondrial content and mitochondrial disease.

**Keywords** autophagy; FBXL4; mitochondrial disease; mtDNA; oxidative phosphorylation

**Subject Categories** Genetics, Gene Therapy & Genetic Disease; Metabolism; Organelles

## Introduction

Mitochondrial diseases are a heterogeneous group of inherited metabolic disorders characterized by deficient oxidative phosphorylation (OXPHOS) that leads to a variety of secondary metabolic defects and pleiotropic clinical phenotypes (Gorman *et al*, 2016). Mutations that cause mitochondrial disorders can be localized to either nuclear DNA or mtDNA and may affect a variety of mitochondrial processes such as mtDNA expression, biogenesis of the OXPHOS system, mitochondrial protein import, or mitochondrial dynamics. Although a given pathogenic mutation often is rare, the collective burden of such mutations is substantial, making mitochondrial diseases the most common form of inherited metabolic disorders.

In order to keep the mitochondrial population of the cell in shape, several quality control mechanisms act on mitochondria. These mechanisms include the cytosolic ubiquitin–proteasome system, several intramitochondrial proteolytic systems, autophagic clearance of damaged mitochondria, and the mitochondria-derived vesicle pathway (Bragoszewski *et al*, 2017; Zimmermann & Reichert, 2017; Pickles *et al*, 2018). Mutations in *Parkin* and *PINK1* cause early-onset forms of Parkinson's disease, and the corresponding proteins have been reported to be involved in mitochondrial protein control (Hauser & Hastings, 2013; Moon & Paek, 2015; Hernandez *et al*, 2016), but their *in vivo* roles remain controversial (Lee *et al*, 2018; Sliter *et al*, 2018). Apart from PARKIN, other ubiquitin ligases, such as FBXO7, MARCH5, and HUWE1, have been related to mitochondrial quality control by directing proteins for proteasomal degradation or by triggering mitophagy (Chen *et al*, 2017; Di Rita *et al*, 2018; Zhou *et al*, 2018). Recently, ubiquitination has also been reported to occur in the inner mitochondrial membrane and this process may possibly contribute to fine-tune metabolism according to the energetic demand of the cell (Lavie *et al*, 2018).

*FBXL4* is a nuclear gene that is implicated in control of mitochondrial function as biallelic mutations recently have been linked to encephalopathy associated with an mtDNA maintenance defect syndrome (Bonnen *et al*, 2013; Gai *et al*, 2013). The disease onset varies from the neonatal period to a few years after birth, and affected patients present a wide range of manifestations, including

1   Department of Medical Biochemistry and Biophysics, Karolinska Institutet, Stockholm, Sweden
2   Max Planck Institute Biology of Ageing - Karolinska Institutet Laboratory, Karolinska Institutet, Stockholm, Sweden
3   Proteomics Core Facility, Max Planck Institute for Biology of Ageing, Cologne, Germnay,
4   Department of Molecular Medicine and Surgery, Karolinska Institutet, Stockholm, Sweden
5   Key Laboratory of Growth Regulation and Translation Research of Zhejiang Province, School of Life Sciences, Westlake University, Hangzhou, China
6   Centre for Inherited Metabolic Diseases, Karolinska University Hospital, Stockholm, Sweden
7   Wellcome Centre for Mitochondrial Research, Translational and Clinical Research Institute, Newcastle University, Newcastle upon Tyne, UK
    *Corresponding author. Tel: +46(0)852483036; E-mail: nils-goran.larsson@ki.se
    †These authors contributed equally to this work

developmental delay, dysmorphology, lactic acidosis, generalized hypotonia, and decreased mtDNA levels (Huemer *et al*, 2015; El-Hattab *et al*, 2017). *FBXL4* mutations are found in ~ 0.7% of all mitochondrial patients and in ~ 14% of children with congenital lactic acidosis, making it one of the most common causes of mitochondrial disease (Dai *et al*, 2017). Despite the high prevalence and severe consequences of pathogenic *FBXL4* mutations, the molecular function of the FBXL4 protein has remained poorly understood (Antoun *et al*, 2015; Huemer *et al*, 2015; Wortmann *et al*, 2015; Dai *et al*, 2017; El-Hattab *et al*, 2017).

FBXL4 belongs to a family of F-box proteins, which typically serve as substrate adaptors in so-called Skp1-cullin1-F-box (SCF) E3 ubiquitin ligase protein complexes (Skaar *et al*, 2013; Zheng *et al*, 2016). However, F-box proteins may also have SCF-independent functions, e.g., as regulators of cellular proliferation, intracellular trafficking, mitochondrial dynamics, and other processes (Nelson *et al*, 2013; Zhou *et al*, 2018). Consequently, SCF-independent functions of FBXL4 cannot be excluded, although it is suggested to be part of an SCF complex (Van Rechem *et al*, 2011).

Previous studies have focused on using patient-derived fibroblasts to characterize the molecular phenotype of *FBXL4* mutations. These studies have shown that loss of FBXL4 is associated with decreased mtDNA levels, decreased levels of OXPHOS protein components, reduced OXPHOS activity, and low oxygen consumption (Bonnen *et al*, 2013; Gai *et al*, 2013), but the precise role of FBXL4 in mitochondria is still unknown. Moreover, cytosolic roles for FBXL4 acting as a component of a ubiquitin ligase complex have been described both in mammalian cell lines and in fruit flies (Van Rechem *et al*, 2011; Li *et al*, 2017).

Here, we describe an *Fbxl4* knockout mouse and show that it recapitulates important phenotypes present in patients with mitochondrial disease caused by *FBXL4* mutations. Using proteomic approaches, we demonstrate that there is a general decrease in mitochondrial proteins accompanied by an increase in lysosomal proteins in *Fbxl4* mouse knockout tissues as well as in fibroblasts derived from patients with loss-of-function *FBXL4* mutations. Surprisingly, expression of nuclear genes encoding mitochondrial proteins and mitochondrial translation remained unaffected in the absence of FBXL4. We present data showing that the molecular phenotype instead is explained by increased autophagic removal of mitochondria, leading to a global decrease in cellular mitochondrial content. Treatment with the lysosomal inhibitor ammonium chloride rescues mitochondrial protein stability in *FBXL4* knockout human cells, consistent with the hypothesis that increased autophagy flux is an important pathophysiological event. In summary,

our results show that increased autophagic removal of mitochondria plays an important role in mitochondrial diseases caused by mutations in *FBXL4*. Further studies are needed to explore the therapeutic potential of these findings, in particular whether inhibition of autophagy may provide a strategy for treatment of affected patients.

# Results

## *Fbxl4* knockout mice have high perinatal mortality

We generated mice with a conditional *Fbxl4* knockout allele and obtained heterozygous knockouts ($Fbxl4^{+/-}$; Fig 1A) after breeding to β-actin-cre mice. We analyzed staged embryos at embryonic day (E) 13.5 and found that $Fbxl4^{-/-}$ embryos were recovered at near the expected Mendelian frequency (Fig 1B) and had a normal gross morphological appearance (Fig 1C). In contrast, very few homozygous knockout ($Fbxl4^{-/-}$) mice were recovered at weaning (Fig 1B) pointing toward late embryonic or perinatal lethality. Chi-square analysis of the data confirmed a significant difference in the genotype distribution in the mice compared with the expected Mendelian ratios ($P$-value < 0.0001). Further analysis showed that there was an increased frequency of dead pups after birth although the exact frequency cannot be determined due to cannibalization by the mother. These findings show that most $Fbxl4^{-/-}$ pups die in the neonatal period.

Patients with *FBXL4* mutations display a significant reduction in mtDNA content (Bonnen *et al*, 2013; Gai *et al*, 2013), and we therefore measured the relative mtDNA copy number by quantitative real-time PCR (qRT–PCR). We found that homozygous knockout ($Fbxl4^{-/-}$) embryos at E13.5 had ~ 25% lower mtDNA content in comparison with wild-type embryos (Fig 1D). However, this moderate reduction in mtDNA copy number is unlikely to affect embryo viability because heterozygous *Tfam* knockout mice are born at Mendelian ratios despite having ~ 50% reduction in mtDNA copy number (Larsson *et al*, 1998).

Surprisingly, the rare surviving $Fbxl4^{-/-}$ animals looked apparently normal until 8–12 months of age when a reduction in body weight was noticed (Fig 1E and F). Most animals (~ 60%) also had a hunch-back appearance at this age (Fig 1E), and ~ 40% of animals had reduced movement, being apathetic or nervous when handled. At 1 year of age, a significant weight difference was observed in both males and females (Fig 1F). The absolute weights of heart, liver, and kidney were normal or near normal in $Fbxl4^{-/-}$ females and males (Fig 1G–I), whereas organ-to-body weight ratios were

---

**Figure 1. *Fbxl4* knockout mice that are viable at the postnatal stage display mild growth defects.**

A   Strategy to generate *Fbxl4* knockout mice. Exon IV was targeted by loxP sites (orange arrows) and removed after Cre recombination. Black arrows indicate Frt sites.

B   Genotype distribution in litters from heterozygous matings ($Fbxl4^{+/-} \times Fbxl4^{+/-}$) at embryonic stage 13.5 (E13.5) and of live animals at weaning. Data are presented as a percentage. Embryos *n* = 79, weaned mice *n* = 339. Dashed lines indicate the expected Mendelian ratios. Chi-square test versus expected Mendelian ratios; ***$P$ < 0.001.

C   Representative images of wild-type (+/+), heterozygous knockout (+/−), and homozygous *Fbxl4* knockout embryos (−/−) at E13.5.

D   Relative levels of mtDNA of *Fbxl4* knockout (−/−) embryos and the corresponding wild-type (+/+) embryos at E13.5. Data represent mean ± SEM, *n* = 5. Student's *t*-test; *$P$ < 0.05.

E   Representative images of male *Fbxl4* knockout (−/−) and matched wild-type (+/+) animals at 1 year of age (left). A hunch-back phenotype was observed both in male and in female knockout animals (right, indicated by arrow).

F–I   Whole body (F), heart (G), liver (H), and kidney (I) weights of 1-year-old animals. Data represent mean ± SEM, *n* ≥ 5 animals. Student's *t*-test; *$P$ < 0.05; **$P$ < 0.01; ***$P$ < 0.001; n.s., not significant ($P$ ≥ 0.05).

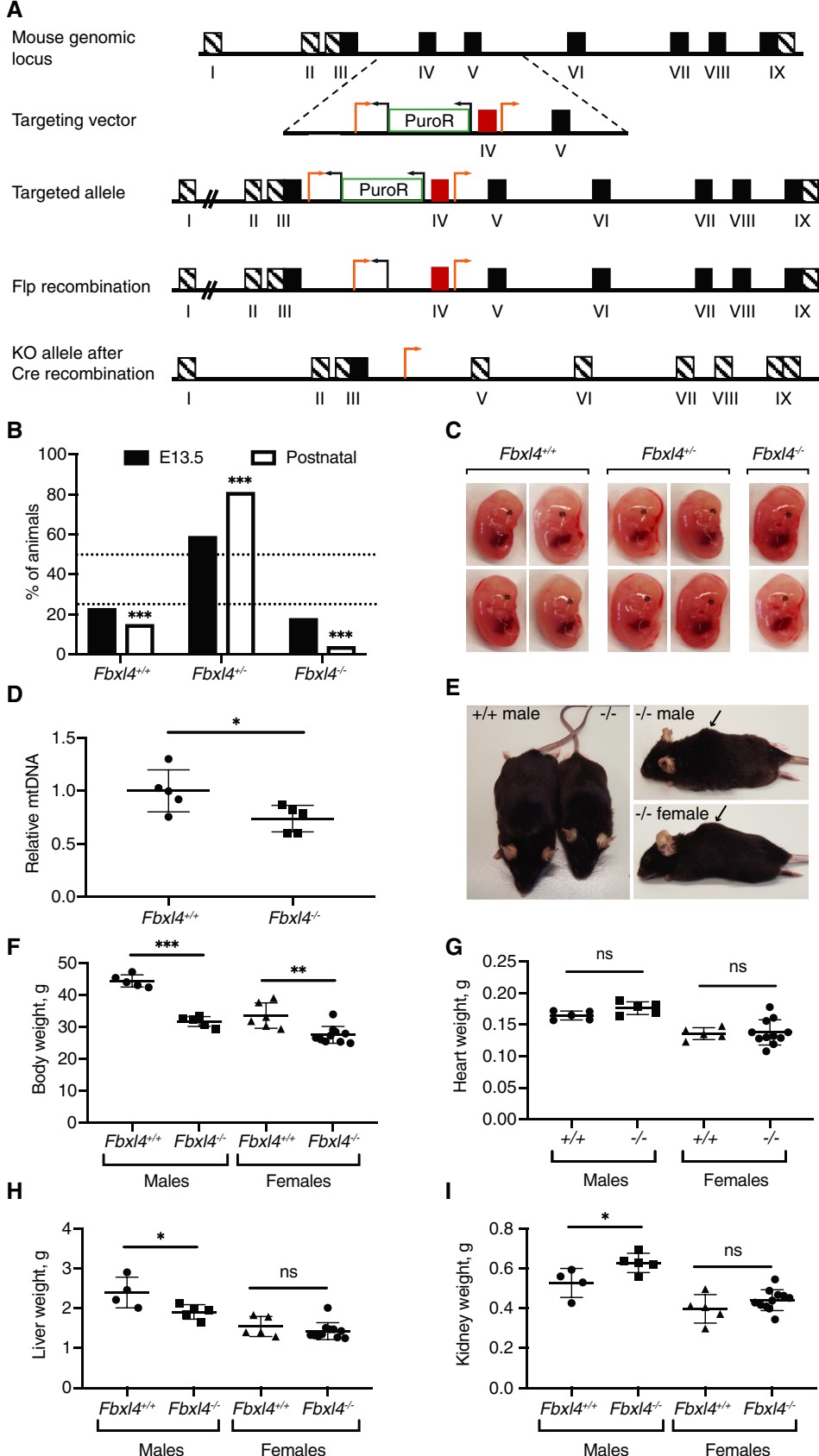

**Figure 1.**

significantly increased for most organs (Fig EV1A–C). Histological examination of organs of 1-year-old $Fbxl4^{-/-}$ animals showed no pathological alterations in the heart nor the kidney (Fig EV1D). However, we observed infiltration of macrophages in the liver, confirmed by Iba1 staining (Fig EV1E), pointing to tissue damage and degeneration. We did not observe differences in COX or sequential COX/SDH reactivities in skeletal muscle sections (Fig EV1F).

### Fbxl4 knockout mice have decreased levels of mtDNA and mitochondrial proteins

Levels of mtDNA were significantly decreased in liver, kidney, and brain but not in heart and skeletal muscle of $Fbxl4^{-/-}$ mice at the age of 1 year (Fig 2A). Also, the mtDNA-encoded transcripts were significantly lower in all analyzed tissues, whereas the levels of nuclear transcripts remained mostly unaffected (Fig 2B).

The decrease in mtDNA and mtDNA-encoded transcripts can be due to a selective depletion of mtDNA or a general decrease in mitochondrial content in the tissue. To investigate this question, we determined the cellular content of mitochondrial proteins and the protein content of isolated mitochondria by using Western blot and mass spectrometry analyses (Figs 2C–F and EV2A–F). The steady-state levels of selected mitochondrial proteins, as determined by Western blotting, were strongly decreased in whole liver homogenates of $Fbxl4^{-/-}$ animals, irrespective of if they were encoded by mtDNA or nuclear DNA (Fig 2C). Tandem mass tag (TMT)-based mass spectrometry analysis of liver tissue homogenate confirmed a global, pathway-independent reduction in mitochondrial proteins, whereas there was an enrichment of lysosomal proteins, such as cathepsins (Ctsb, Ctsc, Ctsd, Ctsl) and other hydrolases (Fig 2E and F, and Dataset EV1). A total of 83 proteins showed significant changes, 57 of which were downregulated and 26 upregulated in homogenate from whole $Fbxl4^{-/-}$ liver (Fig 2E and Dataset EV1). In contrast, label-free, global proteomic analyses of isolated mitochondria from liver, kidney, and heart showed only mild changes in the proteome affecting, e.g., amino acid degradation pathways and propanoate metabolism, consistent with mild metabolic derangement in $Fbxl4^{-/-}$ mitochondria. Importantly, we found no changes in the expression of nuclear-encoded proteins controlling mtDNA maintenance. Analysis of selected proteins by Western blots confirmed the results from proteomics (Fig EV2B, D and F).

The proteomic changes displayed a strong tissue specificity with only six proteins (Pgam5, Grhpr, Acsm5, Spr, Nmat3, and Nit2) consistently changed in mitochondria from all three analyzed tissues (Fig EV2G and Dataset EV2). Kidney and liver, the most affected organs, had 52 commonly changed proteins, mostly belonging to mitochondrial metabolic enzymes and not related to mtDNA maintenance and expression (Dataset EV2).

We proceeded to assess the assembly of OXPHOS complexes in $Fbxl4$ knockouts by resolving respiratory chain complexes from liver mitochondria with blue native polyacrylamide gel electrophoresis (BN–PAGE) followed by in-gel enzyme activity staining for complexes I and IV, and Coomassie staining (Fig EV2H). Despite reduced amounts of the OXPHOS complexes, consistent with reduced steady-state respiratory chain protein levels, no aberrantly assembled or unassembled subcomplexes were observed. In agreement with the BN–PAGE data, spectrophotometric measurement of OXPHOS complex enzyme activities in liver mitochondria did not show major alterations, except a mild, but significant, decrease in complex IV activity (Fig EV2I). Overall, these results are in line with the hypothesis that the mitochondrial dysfunction in $Fbxl4^{-/-}$ animals is predominantly due to reduced cellular mitochondrial content likely through turnover by lysosomes.

### Alterations in patient fibroblasts are similar to the changes in knockout animals

To investigate whether the findings in $Fbxl4^{-/-}$ mice are relevant for the human disease, we characterized fibroblasts obtained from skin biopsies of three patients homozygous for recessive $FBXL4$ mutations (Fig 3A) and three age-matched healthy controls. Patient 1 and Patient 2 have been published previously (Bonnen et al, 2013) and harbor homozygous $FBXL4$ mutations, p.Arg435* and p.Gln519*, respectively (GenBank Accession Number: NM 012160.3). Patient 3 was diagnosed at Karolinska University Hospital and is a compound heterozygote for two independently segregating $FBXL4$ mutations, p.Arg22* on one allele and p.Arg206* on the other allele.

FBXL4-deficient fibroblasts had lower mtDNA content compared with age-matched control fibroblasts (Fig 3B), in line with previously published findings (Bonnen et al, 2013; Gai et al, 2013). The levels of mitochondrial proteins were also significantly lower in patient-derived fibroblasts compared with controls, both for mtDNA- and nuclear-encoded proteins (Fig 3C and D). Importantly,

▶

**Figure 2.** **Fbxl4 knockout mice have reduced amounts of mtDNA and mitochondrial proteins.**

A Relative levels of mtDNA in tissues of 1-year-old $Fbxl4$ knockout mice determined by quantitative real-time PCR (normalized to controls). Data are presented as mean ± SEM, $n = 6$ for controls and $n = 8$ for knockout animals. Student's t-test; **$P < 0.01$; ***$P < 0.001$.

B Transcript levels in indicated tissues of 1-year-old $Fbxl4$ knockout mice determined by quantitative real-time PCR. Relative mRNA levels in control animals were set as 1 (dashed line). Data are presented as mean ± SEM, $n = 6$ for controls and $n = 8$ for knockout animals. Student's t-test; *$P < 0.05$; **$P < 0.01$; ***$P < 0.001$.

C Western blot analysis of protein steady-state levels in liver protein extracts from wild-type ($Fbxl4^{+/+}$) and Fbxl4 knockout ($Fbxl4^{-/-}$) animals.

D Quantification of the Western blot in (C). Signal normalized to the average of controls and presented as mean ± SEM, $n = 4$ for both controls and knockout animals. Student's t-test; **$P < 0.01$; ***$P < 0.001$.

E Quantitative TMT-based proteomic analysis of wild-type and Fbxl4 knockout liver tissue protein extracts. Proteins from three control and three Fbxl4 knockout liver protein extracts were TMT-labeled, quantified, and analyzed separately as described in Materials and Methods; the data from control and knockout animals were averaged after the analysis and presented as $-\log_{10}$ of the adjusted P-value versus $\log_2$ fold change (logFC). Mitochondrial proteins were selected according to mouse MitoCarta 2.0 (Calvo et al, 2016) and highlighted in red; lysosomal proteins were selected according to hLGDB database (Brozzi et al, 2013) and highlighted in yellow.

F Enrichment analysis of lysosomal and mitochondrial proteins according to the logFC median of each group of proteins. Data are presented as 25–75 percentile box with the indicated median, and whiskers representing the ± 1.5× inter-quartile range. Proteins included in each group: total = 1,523, lysosome = 50, mitochondria = 324.

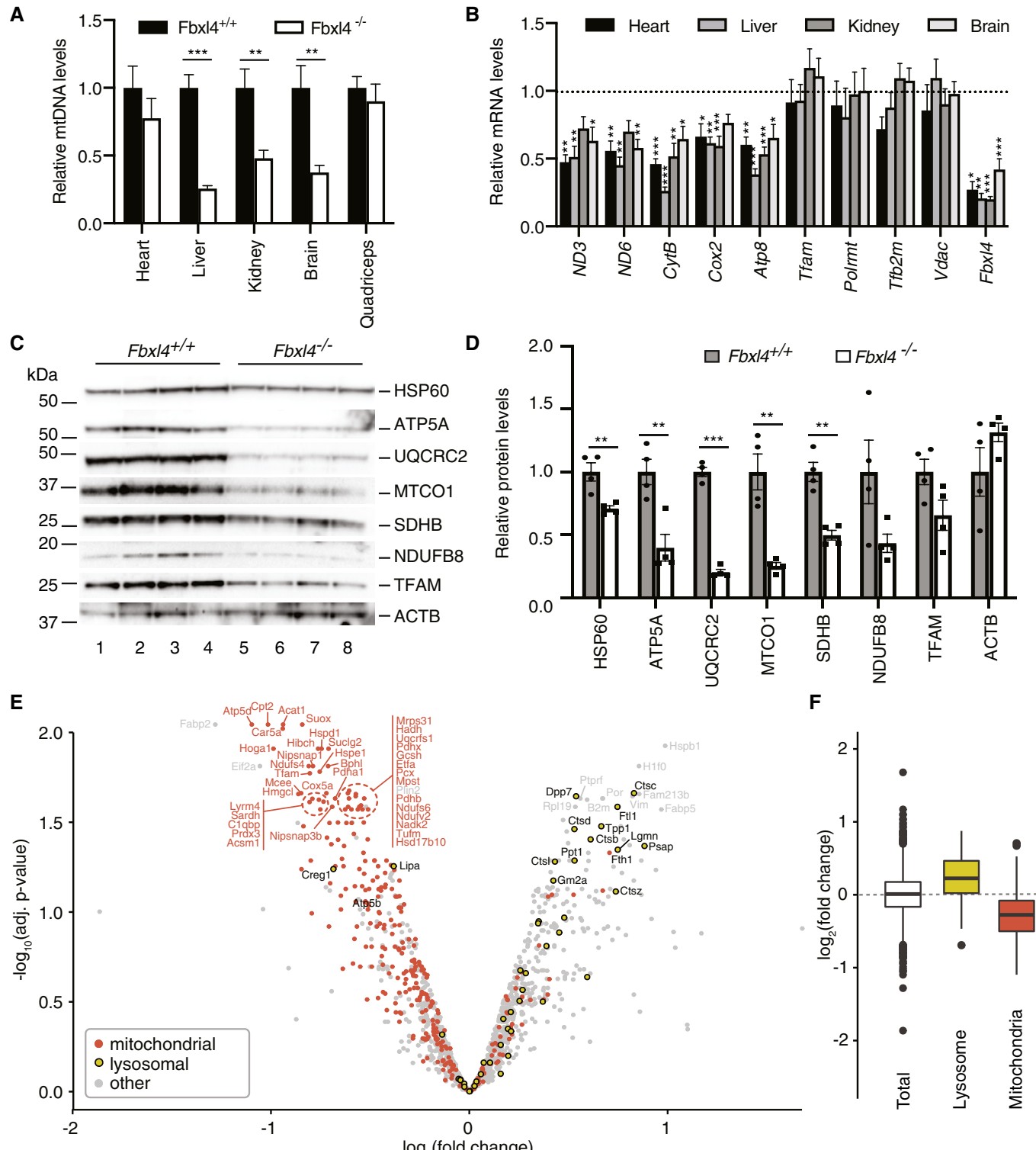

**Figure 2.**

levels of mtDNA-encoded transcripts were decreased, whereas levels of nuclear transcripts encoding mitochondrial proteins were present at normal levels (Fig 3E). The global reduction in mitochondrial protein levels is thus explained by post-transcriptional events rather than by differences in nuclear gene expression.

Tandem mass tag-based quantitative proteomic analysis of patient fibroblasts showed a clear trend with downregulation of many mitochondrial proteins (Fig 3F), which is in good agreement with the Western blot results (Fig 3C). Importantly, FBXL4-deficient fibroblasts have a mild increase in levels of lysosomal proteins

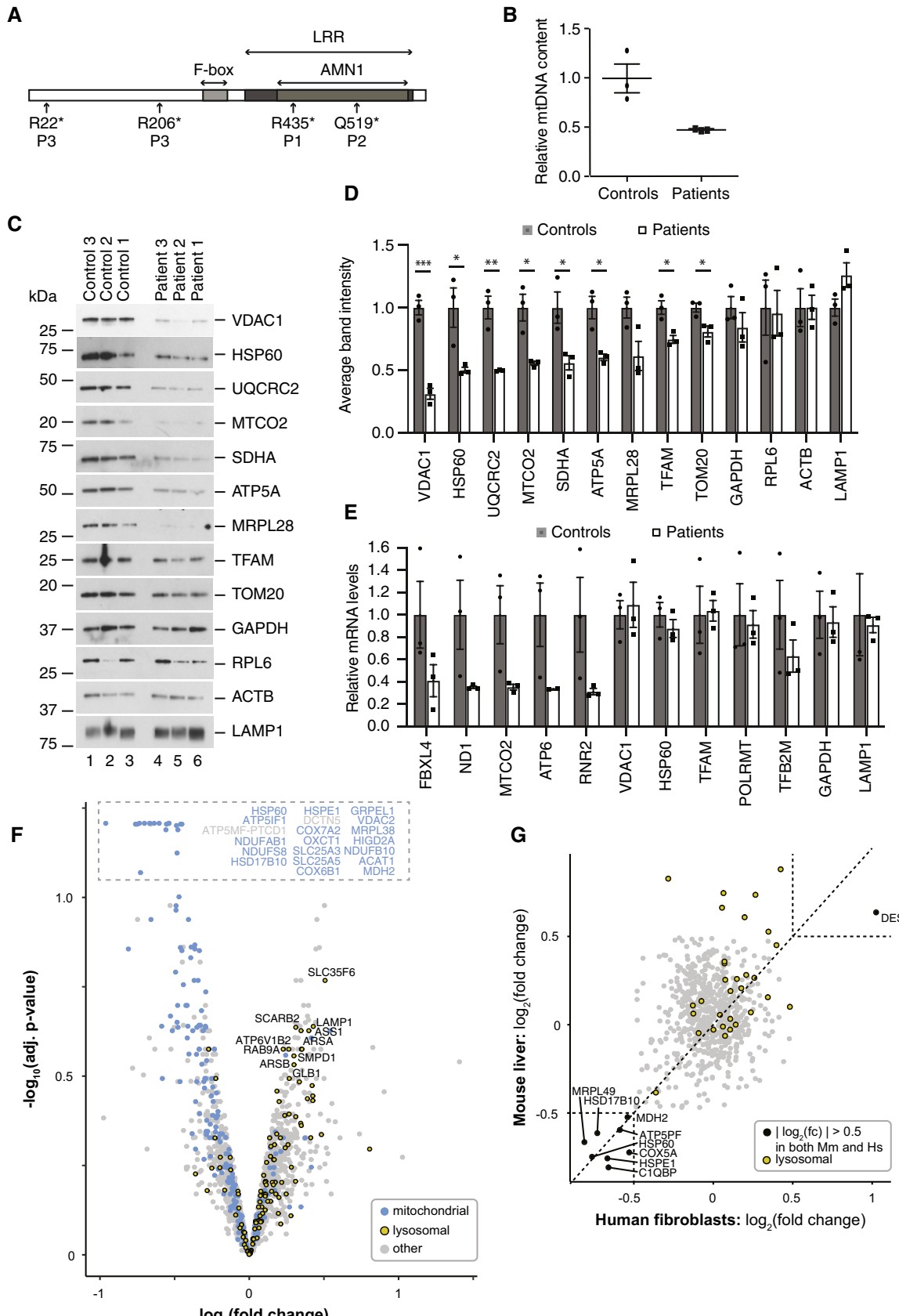

**Figure 3.**

**Figure 3. Mitochondrial alterations observed in FBXL4 patient-derived fibroblasts are consistent with the changes in knockout mice.**

A   Schematic representation of the human FBXL4 protein with amino acid positions of mutations found in patients (P1, P2, and P3). Relative positions of the F-box, leucine-rich repeat (LRR), and antagonist of mitotic exit network protein 1 (AMN1) domains are shown.

B   Relative mtDNA content in fibroblasts from patients and controls as determined by qRT–PCR using an ND6 probe and normalized to nuclear genomic DNA content using an 18S probe. Individual values are presented for each sample, as well as their mean $\pm$ SEM values.

C   Steady-state protein levels in three control and three FBXL4 patient-derived fibroblasts analyzed by Western blotting.

D   Quantification of the Western blot in (C). Signals from the control and FBXL4-mutated cells were considered as biological replicates, normalized to the average of controls, and displayed as mean $\pm$ SEM ($n = 3$ for each). Student's $t$-test; *$P < 0.05$; **$P < 0.01$; ***$P < 0.001$.

E   Quantification of nuclear and mitochondrial transcript levels in control and patient fibroblasts. The results from control and patient cells were averaged, normalized to controls, and displayed as mean $\pm$ SEM ($n = 3$ biological replicates for each condition).

F   Quantitative TMT-based proteomic analysis of control and patient fibroblasts. Proteins from three control and three patient fibroblast cell lines were TMT-labeled, quantified, and analyzed separately as described in Materials and Methods; the data from control and patients were averaged after the analysis and presented as $-\log_{10}$ of the adjusted $P$-value versus $\log_2$ fold change (logFC). Mitochondrial proteins were selected according to human MitoCarta 2.0 (Calvo et al, 2016) and highlighted in blue; lysosomal proteins were selected according to hLGDB database (Brozzi et al, 2013) and highlighted in yellow.

G   $\log_2$FC correlation analysis of mouse liver proteomic dataset and fibroblast proteomic dataset. Proteins with a $\log_2$FC > 0.5 are highlighted in black, and lysosomal proteins are highlighted in yellow.

(Fig 3F and Dataset EV3) similar to the observation in knockout mouse tissues (Fig 2E and F). No alterations in lysosomal morphology were observed in confocal images of Lamp2-stained cells (Fig EV3A). Moreover, we did not observe any significant change in LysoSensor Green DND-189 fluorescence intensity (Fig EV3B). Based on the observation of increased levels of lysosomal proteins in mouse liver and patient fibroblasts lacking FBXL4 (Figs 2E and F, and 3C and D) concomitant with unaffected lysosomal pH (Fig EV3B), we conclude that increased lysosomal mass is increasing mitochondrial turnover.

These findings point toward a conserved mechanism whereby loss of FBXL4 in humans and mice leads to low levels of mitochondrial proteins, as demonstrated by the correlation of both proteomic datasets (Fig 3G). The increase in levels of lysosomal proteins induced by loss of FBXL4 was more marked in mouse tissues than in cultured human fibroblasts (Fig 3G). To summarize, the overall molecular phenotypes in whole tissue/cell extracts of FBXL4-deficient mice and human fibroblasts showed many similarities with low mtDNA levels, low levels of mitochondrial proteins, and increased levels of lysosomal proteins.

### Decreased protein stability in FBXL4 knockout cells

We generated an *FBXL4* knockout RKO cell line by CRISPR/Cas9-mediated gene disruption to get further insights into the role of FBXL4 in a defined cell culture system. The resulting clone had a 20-bp-long deletion in exon 4 of the *FBXL4* gene, leading to a frameshift mutation and a stop codon in the open reading frame (Fig 4A).

The levels of mitochondrial proteins were reduced in the *FBXL4* knockout cell line, similar to the findings in *FBXL4* mutant patient fibroblasts (Fig 4B). To distinguish whether the altered levels of mitochondrial proteins were due to alterations in their biogenesis or degradation, we performed $^{35}$S labeling to study protein synthesis followed by a 16- to 24-h chase to monitor protein stability (Fig 4C). Mitochondrial translation was not affected in the knockout cell line, whereas the amounts of labeled products were significantly reduced after a 24-h chase, indicating an increased turnover of the newly synthesized proteins in the absence of FBXL4.

Given that FBXL4 is an F-box protein suggested to play a role in proteasomal degradation of proteins, we investigated whether inhibition of proteasomal activity could rescue the observed phenotype.

To this end, we performed $^{35}$S labeling of cellular translation products and added the proteasomal inhibitor epoxomicin during the chase phase of the experiment. Epoxomicin treatment increased the differences in protein stability between wild-type and knockout cells and thus promoted increased mitochondrial protein turnover (Fig 4D). In contrast, treatment with the lysosomal inhibitor ammonium chloride rescued the phenotype seen during the chase part of the cellular *in vitro* translation experiment (Fig 4D).

Further analyses showed that LC3-II, p62/SQSTM1, and ubiquitin conjugates accumulated to the same extent in both wild-type and *FBXL4* KO cells after treatment with the lysosomal inhibitor NH$_4$Cl, as well as after treatment with the proteasomal inhibitor epoxomicin (Fig 4E and F).

### Fibroblasts from FBXL4-deficient patients show increased mitophagy

In order to further investigate the increased mitochondrial turnover in the different patient fibroblast lines, we introduced a mito-QC construct using a retroviral vector. Mito-QC is a GFP-mCherry construct targeted to the mitochondria that allow the *in vivo* analysis of mitophagy (Allen *et al*, 2013; McWilliams *et al*, 2016). In the mitochondria, both proteins are fluorescent, but upon internalization in the lysosome the low pH quenches GFP fluorescence but not mCherry.

We analyzed patient fibroblast and a control line using confocal microscopy and observed a mild but significant increase in the number of mitolysosomes/cell area ($\mu m^2$) in the patient lines when compared to the control line expressing mito-QC (Fig 5A and B). These results support the conclusion that pathogenic *FBXL4* variants lead to an increased mitochondrial turnover and, consequently, mitochondrial dysfunction.

Next, we explored the autophagic flux in the patient fibroblast lines by monitoring LC3 conversion and p62 protein levels. We grew the cells in medium without FBS for 3 h and thereafter added NH$_4$Cl to block lysosomal function. As observed in the *FBXL4* KO cells, no differences were found in the LC3-II or p62 levels between control and patient fibroblast lines (Fig EV4A–D). The findings indicate normal autophagic flux and suggest that the increased mitochondrial turnover is dependent on an alternate pathway not involving LC3 conversion. We also checked the accumulation of ubiquitin, p62, and LC3 in sucrose-purified mitochondria (Fig EV4E and F).

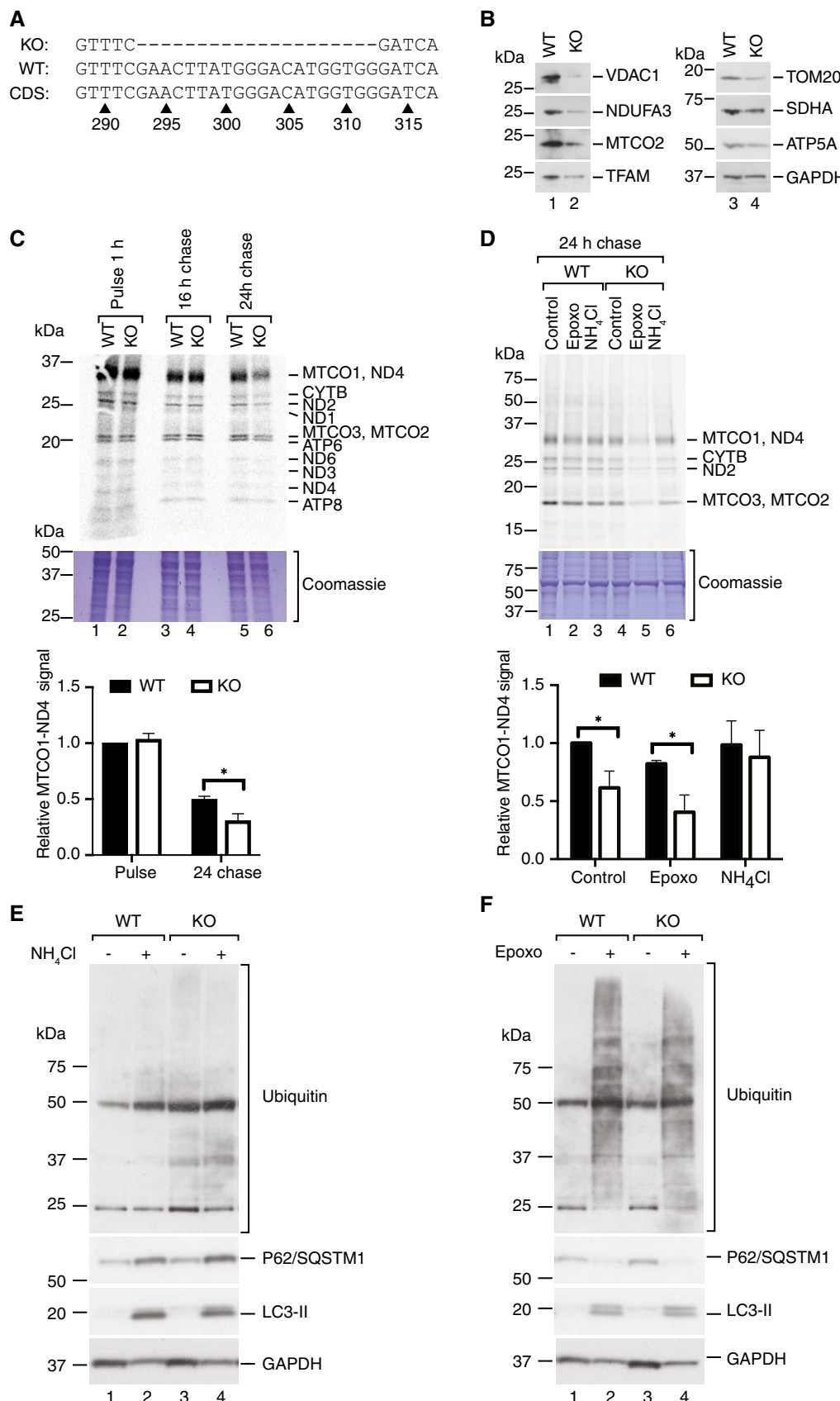

**Figure 4.**

**Figure 4. Lysosomal degradation of mitochondrial proteins is increased in *FBXL4* knockout cells.**

A   Genomic DNA sequence from *FBXL4* knockout (KO) and wild-type (WT) RKO cells was compared to the NCBI reference *FBXL4* coding sequence (CDS) NM_001278716.2 and showed a frameshift-causing deletion in the knockout cells.

B   Steady-state protein levels in wild-type (WT) and *FBXL4* knockout (KO) RKO cells.

C   (Upper panel) Pulse-chase labeling of mitochondrial translation products in wild-type (WT) and FBXL4 knockout (KO) cells. The cells were incubated for 1 h with a $^{35}$S methionine–cysteine labeling mix in the presence of anisomycin to inhibit cytosolic translation (pulse). After removal of the medium, the cells were incubated in complete growth medium without inhibitors and labeled amino acids for 16 or 24 h (chase). A representative experiment of four biological replicates is shown. A Coomassie-stained part of the gel is shown as loading control. (Lower panel) Quantification of the MTCO1-ND4 signal presented as mean $\pm$ SEM; $n$ = 4 biological replicates; Student's $t$-test; *$P$ < 0.05.

D   (Upper panel) Mitochondrial translation products were labeled in wild-type (WT) and FBXL4 knockout (KO) RKO cells for 60 min as in (C), followed by 24-h chase in the presence of 5 $\mu$M epoxomicin (epoxo) or 20 mM ammonium chloride ($NH_4Cl$). A representative experiment of $n \geq 3$ biological replicates is shown. A fragment of the Coomassie-stained gel is shown as loading control. (Lower panel) Quantification of the ND4-MTCO1 signal. $n \geq 3$ biological replicates; data are presented as mean $\pm$ SEM. Student's $t$-test; *$P$ < 0.05.

E, F   Steady-state levels of indicated proteins in wild-type and *FBXL4* knockout RKO cells treated with $NH_4Cl$ (E), or epoxomicin (Epoxo, F).

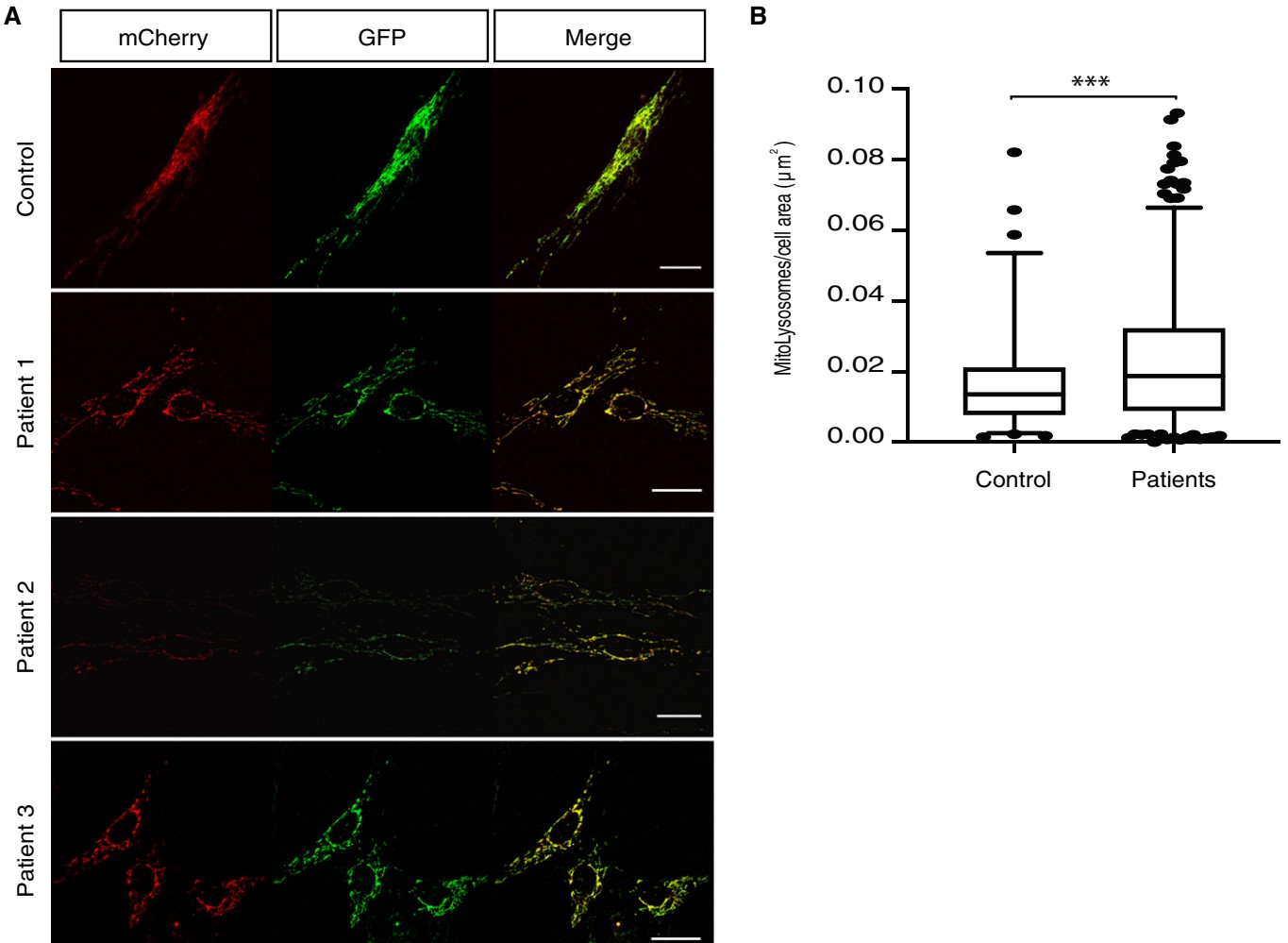

**Figure 5. Mito-QC expression in patient fibroblasts shows increased number of mitolysosomes.**

A   Confocal images from the mito-QC-expressing fibroblast lines. Scale bar 25 $\mu$m.

B   Quantification of the mitolysosomes/cell area ($\mu m^2$) using the mito-QC counter plugin for ImageJ (Montava-Garriga *et al*, 2020). At least three technical replicates for each line and $n$ > 15 cells analyzed for each technical replicate. Data are presented as 25–75 percentile box with the indicated median and whiskers representing 5–95 percentile interval. Non-parametric Mann–Whitney test; ***$P$ < 0.001.

No significant changes were found, supporting the suggestion of activation of an alternate pathway for mitochondrial turnover in the absence of Fbxl4.

In summary, we conclude that the increased mitochondrial protein turnover seen in FBXL4 deficiency is due to increased lysosomal degradation, likely via an alternative autophagy

pathway, and is not explained by increased proteasomal activity. Furthermore, inhibition of lysosomal function rescues the mitochondrial protein stability phenotype in FBXL4 deficiency, suggesting that decreasing autophagy may be a strategy for the treatment of affected patients.

## Discussion

The pathophysiology of human FBXL4 deficiency causing an encephalopathic syndrome associated with mtDNA depletion has remained enigmatic. Here, we present evidence that FBXL4 acts as a regulator preventing excessive turnover of mitochondria via lysosomes. We have characterized *Fbxl4* knockout mice, FBXL4-deficient patient fibroblast, and a human *FBXL4* knockout cell line and report several lines of evidence showing that loss of FBXL4 leads to increased mitochondrial turnover via lysosomes.

Quantitative proteomics showed an overall decrease in mitochondrial content and an increase in lysosomal proteins both in mouse tissues and in human fibroblasts lacking FBXL4. These results led us to hypothesize that mitochondrial content was decreased due to increased lysosomal degradation. This assumption was supported by $^{35}$S labeling experiments, which showed that mitochondrial protein stability in *FBXL4* knockout cells was reduced but could be rescued by inhibiting lysosomal hydrolases with ammonium chloride. In further support of the hypothesis, the mito-QC expression in patient fibroblasts showed an increased number of mitolysosomes consistent with increased mitochondrial turnover. However, the levels of LC3-II and also p62, a global autophagy marker (Calvo-Garrido *et al*, 2019), were similar in wild-type and *FBXL4* knockout cells, also there was no difference in levels of these proteins in control and *FBXL4* mutant human fibroblast lines. This finding can be explained by an alternative autophagy pathway that is independent of LC3. Such a pathway has been reported in Atg5 knockout mice and is dependent on the Rab GTPase, Rab9a (Kuma *et al*, 2004). This pathway involves the participation of vesicles from the trans-Golgi that engulf the cargos for subsequent fusion with lysosomes (Kuma *et al*, 2004). There is increasing evidence that Rab GTPases can play an important role during autophagy and several members of this protein family have been related to different steps in the autophagy process (Ao *et al*, 2014). Interestingly, a slight increase in different Rab GTPases was observed in both mouse and human fibroblasts lacking FBXL4, including Rab9a in the case of human fibroblasts, which lends support to the hypothesis of the existence of an alternate autophagy pathway driven by this group of proteins. Another option that cannot be excluded is a role for the MDV pathway which is also independent of LC3 conversion (Sugiura *et al*, 2014).

As we did not observe changes in the conversion of LC3-I to LC3-II, we assume that the classical pathway for autophagic flux in FBXL4-deficient cells is not affected. Instead, an alternate autophagy pathway may be upregulated leading to increased mitochondrial removal. This model seemingly contradicts the proposed role of FBXL4 as an F-box protein in the SCF complex. One possible explanation for this apparent discrepancy implies an important role for FBXL4 in mitochondrial quality control as a substrate-recognizing subunit of the ubiquitin ligase complex.

Consistent with this model, the absence of FBXL4 would lead to accumulation of aberrant mitochondrial proteins that could affect mitochondrial function and thus increase the degradation of mitochondria by the lysosome. However, there is no evidence showing the presence of the SCF complex in mitochondria and it is therefore unlikely that ubiquitination could happen in the mitochondrial intermembrane space (IMS) via FBXL4. Recently, it has been reported that ubiquitination could occur in the inner mitochondrial membrane, but the ubiquitin ligases or complexes catalyzing this reaction have not been identified (Lavie *et al*, 2018). Another possibility, although perhaps also unlikely, is that FBXL4 could recognize unfolded or damaged proteins in the IMS and promote their export to the cytosol for degradation. A somewhat similar mechanism has been shown in yeast, in which unfolded proteins can be retro-translocated from the IMS to the cytosol through the TOM complex and further degraded by the ubiquitin–proteasomal system in the cytosol (Bragoszewski *et al*, 2015).

The phenotypes of *Fbxl4*$^{-/-}$ mice agree with the alternate autophagy pathway model. The finding that *Fbxl4*$^{-/-}$ embryos are present at Mendelian ratios and develop normally until E13.5 despite having somewhat lower mtDNA levels argues that the mitochondrial phenotype is not the leading cause of the perinatal death. It should be pointed out that neonatal lethality has been reported in many autophagy-deficient animals, e.g., *Atg3*, *Atg5*, *Atg7*, and *Atg12* knockouts (Kuma *et al*, 2004, 2017; Mizushima & Levine, 2010). Interestingly, *Atg4b* knockout mice were reported to be born at non-Mendelian ratios, but developed almost normally after birth, demonstrating a striking similarity to the *Fbxl4*$^{-/-}$ phenotype (Read *et al*, 2011). Although *Fbxl4* knockout animals are not autophagy-deficient, we assume that increased mitophagy flux at birth could imbalance the overall autophagic states of the cell leading to mitochondrial deficiency. The decreased mitochondrial content caused by increased autophagic removal likely has a particularly deleterious impact in the neonatal period as it may prevent the crucial metabolic switch from glycolysis to OXPHOS in newborn animals. The few newborn FBXL4-deficient animals that survive this critical period will develop a slowly progressive mitochondrial phenotype much later in life.

In summary, the results we present here argue that the molecular phenotypes in FBXL4-deficient patients are explained by a global decrease in mitochondrial content and not by a specific role for FBXL4 in mtDNA maintenance or biogenesis of the OXPHOS complexes. Importantly, the finding that inhibition of lysosomal function can rescue molecular phenotypes in FBXL4-deficient cells suggests that manipulation of mitochondrial clearance by autophagy may be utilized as a future strategy to develop treatments of affected patients.

## Materials and Methods

### Fbxl4 knockout mice

*Fbxl4* knockout mice were generated by Taconic Artemis by targeting the gene in embryonic stem cells derived from C57BL/6N using a vector from the C57BL/6J RPCI-23 BAC library. To generate the *Fbxl4* knockout allele, exon 4 was flanked by loxP sites, and a

puromycin resistance (PuroR) cassette flanked by Frt sites was also introduced as a selection marker. The puromycin resistance cassette was removed by crossing *Fbxl4*$^{+/loxP-Puro}$ mice with mice expressing Flp recombinase. *Fbxl4*$^{+/loxP}$ mice were thereafter mated with mice ubiquitously expressing Cre recombinase (β-actin-cre) to generate heterozygous *Fbxl4* knockouts. Finally, intercrosses of heterozygous knockout animals were performed to generate *Fbxl4* knockout animals (*Fbxl4*$^{-/-}$). Animals were housed in standard ventilated cages at 12-h light/dark cycle and fed with a normal chow diet *ad libitum*. For isolation of tissues, animals were sacrificed by cervical dislocation and dissected immediately. Isolated organs were snap-frozen in liquid nitrogen and stored at −80°C. Animal studies were approved by the animal welfare ethics committee and performed in compliance with European law.

## Histology

For heart, kidney, and liver microscopy analyses, tissues were fixed in 4% PFA, dehydrated, and paraffin-embedded. Sections of 5 μm in thickness were mounted on glass slides, de-paraffinized with xylene, and stained with hematoxylin and eosin.

## Immunohistochemical analysis

Liver sections were de-paraffinized with xylene and rehydrated by successive incubations in ethanol 100, 95, and 70% and water. Thereafter, slides were incubated in 10 mM sodium citrate buffer (pH 6) at sub-boiling temperature for 10 min and cooled at room temperature for 30 min. Before staining, slides were washed three times with water and incubated in 3% hydrogen peroxide for 10 min to quench endogenous peroxidase activity and washed twice in water. Thereafter, slides were blocked in 5% goat serum in TBST. Iba1 primary antibody (Wako) was used to detect Iba1 expression. Later, we used the VectaStain ABC HRP Kit (Vector Labs) and diaminobenzidine to detect immunoreactivity. Finally, slides were counter-stained with hematoxylin, dehydrated, and mounted for bright-field microscopy.

## Sequential COX/SDH histochemistry

COX/SDH histochemistry was performed as previously described (Filograna *et al*, 2019). Briefly, quadriceps from wild-type and *FBXL4* knockout animals were washed in PBS and frozen in liquid nitrogen-cooled isopentane. Sections, 14 μm, were mounted in poly-lysine-coated glass slides and left air-dry briefly. Sections were incubated for 45 min at 37°C in COX medium (100 μM cytochrome c, 4 mM diaminobenzidine tetrahydrochloride, catalase (20 μg/mL), and 0.2 M phosphate buffer [pH 7]). Next, slides were washed in PBS and incubated for 30 min at 37°C in SDH medium (130 mM sodium succinate, 200 μM phenazinemethosulphate, 1 mM sodium azide, 1.5 mM nitroblue tetrazolium, and 0.2 M phosphate buffer [pH 7]). Finally, slides were washed in PBS, dehydrated in increasing ethanol concentrations, and mounted for bright-field microscopy.

## RNA extraction and gene expression analysis

RNA from mouse tissues or cells was extracted using TRIzol Reagent (Invitrogen) following the manufacturer's instructions. Total RNA

(1 μg) was converted to cDNA using the High-Capacity cDNA Reverse Transcription Kit (Applied Biosystems), and 10 ng of cDNA was used in each qPCR. These reactions were done with TaqMan Universal MasterMix (Applied Biosystems) in a QuantStudio 6 Flex (Applied Biosystems). Specific primers and probes were purchased from Applied Biosystems (TaqMan gene expression assays). The acquired data were analyzed with QuantStudio Real-Time PCR Software (v1.1). Relative expression was calculated based on Ct values for each gene normalized with the values obtained for housekeeping genes (Actb, B2M, or 18S ribosomal RNA). All probes used in this study are listed in Appendix Table S1.

## DNA extraction and mtDNA quantification

Total DNA was isolated with standard phenol/chloroform and proteinase K digestion protocol, and the concentration was measured with Qubit DNA assay (Thermo Fisher Scientific). Quantitative PCR was performed to quantify mtDNA, using 5 ng of total DNA in each reaction. Specific probes for *ND1* (mouse) or *ND6* (human) were used to detect mtDNA, whereas the *Actb* probe (for mouse) or the 18S ribosomal RNA probe (for human DNA) was used to detect nuclear DNA. The relative mtDNA levels were calculated as the ratio of mtDNA/nDNA based on the obtained Ct values.

## Cell culture

Mammalian cells were cultured in DMEM GlutaMax (Invitrogen) medium supplemented with 10% fetal bovine serum (FBS) and 1× PenStrep. Cells were grown in an incubator at 37°C in a 5% $CO_2$ atmosphere. Patient fibroblasts lines 1 (p.Arg435*) and 2 (p.Gln519*) both harbor pathogenic, homozygous loss-of-function *FBXL4* mutations and were previously described in Bonnen *et al*, 2013 (S2 and S1, respectively, in the original report) (Bonnen *et al*, 2013). Patient fibroblast line 3 was obtained from a patient diagnosed at Karolinska University Hospital. Whole-genome sequencing of this patient identified two previously unreported heterozygous independently segregating stop mutations c.64C>T, p.Arg22* and c.616C>T, p.Arg206* in the *FBXL4* gene. The patient had widened lateral ventricles of the brain on intrauterine ultrasound screening, but showed no symptoms until 6 months of age when delayed motor development and general hypotonia became evident. At this age, the patient also had hyperlactatemia (p-lactate 6.5 mmol/l), hypochromic anemia, and neutropenia. Organic acids in urine showed elevated lactate, fumarate, and malate. There was decreased ATP synthesis, and the activities of OXPHOS complexes I, I + III, II, II + III and IV were reduced in isolated muscle mitochondria. Control fibroblasts used in this study were obtained from age-matched healthy individuals. Informed consent for research studies was obtained for all subjects or their parents and for controls in accordance with the Declaration of Helsinki protocols, the Department of Health and Human Services Belmont Report, and approved by the Regional Ethics Committee at Karolinska Institutet in Stockholm, Sweden.

The *FBXL4* CRISPR knockout cell line was generated as previously described (Ran *et al*, 2013). Briefly, a DNA sequence encoding a gRNA targeting *FBXL4* exon 4 was cloned into the pX459v2 vector. The RKO cell line (ATCC) was transfected with this vector using Lipofectamine 3000 (Thermo Fisher Scientific) following the

manufacturer's recommendations. Transfected cells were selected with puromycin, and obtained clones were sequenced to analyze disruption of the *FBXL4* gene.

## Immunofluorescence

For Lamp2 immunofluorescence, cells were grown on glass coverslips. Coverslips were washed twice with PBS and fixed with 4% PFA in PBS for 15 min. After, cells were incubated for 1 h in 1% BSA, and 0.1% saponin in PBS. Cells were then incubated with Lamp2 antibody (Abcam) in 1% BSA, and 0.1% saponin in PBS at 4°C overnight. Coverslips were washed two times with PBS and then incubated with Alexa 568 anti-mouse secondary antibody for 1 h at room temperature. Finally, coverslips were stained with DAPI, washed five times with PBS, and mounted using Prolong Diamond Antifade Mountant (Life Technologies). The slides were imaged in a Zeiss LSM700 confocal microscope.

## LysoSensor Green fluorescence measurement

A total of 1,000 cells were plated in black p96 well plates and allowed to attach. Thereafter, the medium was replaced with DMEM with 2 μM LysoSensor Green DND-189 (Life Technologies) and cells were incubated at 37°C for 30 min. Finally, cells were washed twice with PBS and fluorescence was immediately measured in a Tecan Infinite M200 Pro (Tecan). A total of five technical replicates were done for each line, and in each technical replicate, at least three wells per line were measured.

## Mito-QC expression, imaging, and analysis

Mito-QC pBabe vector was kindly provided by Dr. Ian Ganley. Transfection of fibroblast lines was done as previously described (Allen *et al*, 2013). Cells stably expressing mito-QC were plated on glass coverslips. Coverslips were washed with PBS and fixed with 4% PFA, and 200 mM HEPES-KOH pH = 7 for 15 min and washed twice with PBS. Coverslips were mounted using Prolong Diamond Antifade Mountant (Life Technologies) and imaged in a Zeiss LSM700 confocal microscope. Obtained images were analyzed using the ImageJ mito-QC counter semi-automated tool (Montava-Garriga *et al*, 2020). Ratio threshold was set to 0.6, radius for smoothing = 1, and standard deviation over mCherry mean intensity = 3. At least three technical replicates were done for each line and > 15 cells analyzed in each replicate.

## Preparation of pure mitochondrial fractions

Pure mitochondrial fractions were obtained as previously reported (Kühl *et al*, 2017). Briefly, crude mitochondrial fractions were isolated from mouse liver, kidney, and heart using differential centrifugation procedures. The crude mitochondrial pellets were washed in 1xM buffer containing 220 mM mannitol, 70 mM sucrose, 5 mM HEPES ph = 7.4, and 1 mM EGTA, and thereafter, mitochondria were purified in a Percoll gradient (12–19–40%) through centrifugation in a SW41 rotor at 42,000 *g* at 4°C for 30 min in a Beckman Coulter Optima L-100 XP ultracentrifuge. Mitochondria were collected between 19 and 40% phases, and washed three times with 1xM buffer. Mitochondrial pellets were frozen at −80°C.

## SDS–PAGE, BN–PAGE, and Western blotting

The protein content of samples was measured by Pierce™ BCA Protein Assay Kit (Thermo Fisher Scientific), and equal protein amounts were loaded on a gel. For SDS–PAGE, precast Bolt Bis-Tris Plus Gels (Thermo Fisher Scientific) were used according to manufacturer's instructions. For Western blot analysis, the gels were blotted and membranes probed with antibodies of interest. Signal detection was done using HRP-coupled secondary antibodies with enhanced chemiluminescence substrate (Bio-Rad). A list of the used primary antibodies is provided in Appendix Table S2.

BN–PAGE was performed using the NativePAGE system (Thermo Fisher Scientific). In brief, 75 μg of mitochondria was solubilized in solubilization buffer: 1% (w/v) digitonin, 20 mM Tris–HCl pH 7.4, 0.1 mM EDTA, 50 mM NaCl, and 10% glycerol (v/v). After 10 min of incubation on ice, samples were centrifuged to eliminate non-solubilized material and supernatant was mixed with 5% w/v Coomassie Brilliant Blue G-250. Samples were separated in Native-PAGE 3–12% Bis-Tris gels (Thermo Fisher Scientific) followed by staining with Imperial Staining (Thermo Fisher Scientific) or in-gel complex activity staining as described previously (Matic *et al*, 2018). For complex I staining, gels were soaked in Tris–HCl pH 7.4 buffer containing 0.1 mg/ml NADH and 2.5 mg/ml iodotetrazolium chloride. For complex IV staining, gels were soaked in 50 mM phosphate buffer pH 7.4 with 0.5 M sucrose, 0.5 mg/ml 3,3′-diamino-benzidine tetrahydrochloride, 1 mg/ml cytochrome *c*, and 20 μg/ml catalase.

## Biochemical assessment of OXPHOS enzyme activities

For the determination of OXPHOS biochemical activities, we used pure mitochondria isolated from liver. Protein concentration from Percoll purified mitochondria fractions was determined using Pierce™ BCA Protein Assay Kit (Thermo Fisher Scientific) and resuspended in a buffer containing 250 mM sucrose, 15 mM magnesium acetate, 2 mM EDTA, 0.5 g/l BSA, and 15 mM potassium dihydrogen phosphate pH = 7.2. Biochemical activities of the respiratory chain complexes were determined spectrophotometrically with standard methods as previously described (Wibom *et al*, 2002).

## $^{35}$S labeling of translation products in cells

Labeling of mitochondrial translation products with $^{35}$S-methionine was performed as described previously (Chomyn, 1996). Cells were grown in 24-well plates, extensively washed with PBS, and pre-incubated in Cys-Met-free DMEM medium (Gibco) for 20 min. Next, the cells were treated for 20 min with 100 μg/ml anisomycin (Sigma) to block cytosolic translation, and 3.7 MBq of $^{35}$S-L-methionine and cysteine mix (PerkinElmer) was added to the cells for 60 min to label newly synthesized mitochondrial translation products. For the chase experiment, cells were extensively washed with PBS and complete DMEM + FBS medium to remove radioactive substances and anisomycin, and then grown for 16–24 h. The radiolabelled translation products were resolved by SDS–PAGE, and the gels were dried and exposed to phosphor screens for radioactive signal detection.

### Protein extraction, proteolytic digestion, and chemical labeling

Cell pellet samples were suspended in a mixture of 0.1% ProteaseMax (Promega), 4 M urea (Sigma-Aldrich), 50 mM ammonium bicarbonate, and 10% acetonitrile (AcN), and were sonicated with a VibraCell probe (Sonics & Materials, Inc.) for 1 min, with pulse 2/2, at 20% amplitude. Following sonication in a water bath for 5 min, extracts were vortexed and centrifuged for 5 min at 16,000 g. The supernatants were thereafter transferred to new tubes, and protein concentrations were determined in a 1:10 dilution with water yielding 500–1,300 μg protein (mouse fibroblasts).

Tryptic digestion with an enzyme-to-protein ratio of 1:50 was performed with 25 μg of each sample using a protocol including reduction in 6 mM dithiothreitol at 37°C for 60 min, alkylation in 22 mM iodoacetamide for 30 min at room temperature in the dark, and proteolysis over night at 37°C. Tryptic peptides were cleaned with C18 HyperSep Filter Plate, bed volume 40 μl (Thermo Fisher Scientific), and dried on a SpeedVac (miVac, Thermo Fisher Scientific). Six of TMT10plex reagents (Thermo Fisher Scientific) in 100 μg aliquots were dissolved in 30 μl dry AcN, scrambled, and mixed with the digested samples dissolved in 70 μl TEAB (resulting final 30% AcN), followed by incubation at 22°C for 2 h at 550 rpm. The reaction was then quenched with 12 μl of 5% hydroxylamine at 22°C for 15 min at 550 rpm. Finally, the labeled samples were pooled and dried in a SpeedVac.

Enriched mitochondria were lysed in 6 M guanidinium chloride, 10 mM Tris–HCl, 40 mM chloroacetamide, and 100 mM Tris–HCl by heating the solution at 95°C for 10 min, followed by repeated cycles of sonication for 10 and 30 s and a 30-second pause on a Bioruptor (Diagenode). Samples were centrifuged for 20 min at 20,000 g, and the supernatant was diluted 1:10 using 20 mM Tris–HCl pH 8. Overnight digestion was done using 1 μg Trypsin Gold (Promega) at 37°C. The supernatant was acidified with 50% formic acid, and the solution was desalted using home-made StageTips (Rappsilber *et al*, 2003).

### Liquid chromatography–tandem mass spectrometry

Samples, 2 μg aliquots, were injected into a 50-cm-long C18 EASY spray column (Thermo Fisher Scientific) installed in a nano-LC-1000 system online coupled to a Orbitrap Fusion mass spectrometer (Thermo Fisher Scientific, Bremen, Germany). The chromatographic separation of the peptides was achieved with the organic gradient as follows: 2–26% AcN in 110 min, 26–35% AcN in 10 min, 35–95% AcN in 5 min, and 95% AcN for 15 min at a flow rate of 300 nl/min. Full mass scans were acquired in $m/z$ 375–1,500 at resolution of $R = 120,000$ (at $m/z$ 200), followed by data-dependent HCD fragmentations from maximum 15 most intense precursor ions with a charge state 2+ to 7+. The tandem mass scans were acquired with a resolution of $R = 60,000$, targeting $5 \times 10^4$ ions, setting isolation width to $m/z$ 1.4, and normalized collision energy to 35%.

For label-free proteomic analysis, peptides were separated on a 25-cm, 75-μm internal diameter PicoFrit analytical column (New Objective) packed with 1.9 μm ReproSil-Pur 120 C18-AQ media (Dr. Maisch,) using an EASY-nLC 1200 (Thermo Fisher Scientific). The column was maintained at 50°C. Buffer A and buffer B were 0.1% formic acid in water and 0.1% formic acid in 80% acetonitrile. Peptides were separated on a segmented gradient from 6 to 25%

buffer B for 135 min, from 25 to 31% for 20 min, and from 31 to 50% buffer B for 20 min at 200 nl/min. Eluting peptides were analyzed on a QExactive HF (Thermo Fisher Scientific). Peptide precursor $m/z$ measurements were carried out at 60,000 resolution in the 300–1,800 $m/z$ range. The ten most intense precursors with charge state from 2 to 7 only were selected for HCD fragmentation using 25% normalized collision energy. The $m/z$ values of the peptide fragments were measured in the Orbitrap at 30,000 resolution, using a minimum AGC target of 8e3 and 55-ms maximum injection time. Upon fragmentation, precursors were put on a dynamic exclusion list for 45 s.

### Protein identification and quantification

Acquired raw data files were loaded in Proteome Discoverer v2.2 and searched against mouse or human SwissProt protein databases (42,793 and 21,008 entries, respectively) using the Mascot 2.5.1 search engine (Matrix Science Ltd.). A maximum of two missed cleavage sites were allowed for trypsin, while setting the precursor and the fragment ion mass tolerance to 10 ppm and 0.05, respectively. Dynamic modifications of oxidation on methionine, deamidation of asparagine and glutamine, and acetylation of N-termini were set. For quantification, both unique and razor peptides were requested. The final quantitative data analysis was performed with an in-house developed R script.

Label-free proteomic data were analyzed with MaxQuant version 1.6.1.0 (Cox & Mann, 2008) using the integrated Andromeda search engine (Cox *et al*, 2011). Peptide fragmentation spectra were searched against the canonical and isoform sequences of the mouse reference proteome (proteome ID UP000000589, downloaded in September 2018 from UniProt). Methionine oxidation and protein N-terminal acetylation were set as variable modifications; cysteine carbamidomethylation was set as fixed modification. The digestion parameters were set to "specific" and "Trypsin/P". The minimum number of peptides and razor peptides for protein identification was 1; the minimum number of unique peptides was 0. Protein identification was performed at a peptide spectrum matches and protein false discovery rate of 0.01. The "second peptide" option was on. Successful identifications were transferred between the different raw files using the "Match between runs" option. Label-free quantification (LFQ) (Cox *et al*, 2014) was performed using an LFQ minimum ratio count of two. LFQ intensities were filtered for at least two valid values in at least one genotype, and missing values were imputed from a normal distribution with a width of 0.3 and downshift of 1.8. Analysis of the label-free quantification results was performed using R (R Development Core Team, 2010). MitoCarta annotations were added using the corresponding Gene names (primary) entry and the first of the Gene names (synonym) entries of the oldest UniProt ID with the highest number of peptides in the column "Protein IDs". Only mitochondrial proteins, according to MitoCarta, were used for analysis. Protein label-free intensities were normalized using vsn (Huber *et al*, 2002). Differential expression analysis was performed using the two-sided moderated *t*-test from the limma package (Ritchie *et al*, 2015). Exploratory data analysis was performed using ggplot and GGally.

### Statistical methods

Data were statistically analyzed using GraphPad Prism 8 software (except proteomics data as detailed above). Neither randomization

nor blinding was implemented for animal studies and data analysis. All data are presented by means ± SEM. When two groups were compared, a two-sided unpaired Student's *t*-test was used, while a one-way ANOVA and Tukey's *post hoc* analysis were used for multiple group comparison. Mito-QC data are presented as 5–95% percentile boxplot, and non-parametric Mann–Whitney test was used for statistical analysis. Genotype distribution in mice and embryos was analyzed using chi-square test by comparing the obtained percentages against the expected Mendelian percentages. *P*-values < 0.05 were considered statistically significant. The exact number of replicates and *P*-values for each figure are summarized in Appendix Table S3.

## Data and software availability

The datasets produced in this study are available in the following databases: Mass spectrometry datasets have been deposited to the ProteomeXchange Consortium (Perez-Riverol *et al*, 2019): PRIDE PXD18639 (http://www.ebi.ac.uk/pride/archive/projects/PXD18639).

**Expanded View** for this article is available online.

## Acknowledgements

This study was supported by grants to NGL from the Swedish Research Council (2015-00418), the Knut and Alice Wallenberg Foundation, the European Research Council (Advanced Grant 2016-741366), the Swedish Cancer Society (2018.602), and the Swedish state under the agreement between the Swedish government and the county councils, the ALF agreement (SLL2018.0471). AWr was supported by the Swedish Research Council (VR2016-02179). RWT is supported by the Wellcome Centre for Mitochondrial Research (203105/Z/16/Z), the Medical Research Council (MRC) International Centre for Genomic Medicine in Neuromuscular Disease, the Mitochondrial Disease Patient Cohort (UK) (G0800674), the Lily Foundation, the UK NIHR Biomedical Research Centre for Ageing and Age-related disease award to the Newcastle upon Tyne Foundation Hospitals NHS Trust, the MRC/EPSRC Molecular Pathology Node, and the UK NHS Highly Specialised Service for Rare Mitochondrial Disorders of Adults and Children (http://www.newcastle-mitochondria.com/). Mass spectrometric analyses and database searches for protein identification and quantification in mouse tissues and cultured cells were carried out at the Proteomics Biomedicum core facility, Karolinska Institutet, Stockholm, with an outstanding support by Dr. Akos Vegvari. We would like to thank Avan Taha and Dr. Xinping Li for performing proteomic sample preparation of mouse mitochondria and mass spectrometric measurements at the proteomics core facility at the Max Planck Institute for Biology of Ageing, Cologne, Germany. We thank Christian Kukat, Alexandra Just, and Anna-Lena Schumacher of FACS and Imaging Core Facility at Max Planck Institute for their assistance with histology experiments. We thank Rolf Wibom for measurements of respiratory chain enzyme activities. We thank Dr. Ian Ganley for the kind gift of the pBabe mito-QC vector. The authors are grateful to Prof. Nico Dantuma, Dr. Olle Sangfelt, and Dr. Javier Calvo-Garrido for productive discussions and useful advice.

## Author contributions

DA, OL, and N-GL conceived the project, designed the experiments, and wrote the manuscript. DA and OL performed and interpreted the majority of the experiments. DA, OL, CK, and N-GL advised on methodology and interpreted the data. AS, IA, and MJ performed experiments and analyzed the data. FS contributed to proteomic data analysis. AWe, RWT, and AWr provided critical input for the project. N-GL supervised the project. All the authors commented on the manuscript.

## Conflict of interest

The authors declare that they have no conflict of interest.

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

**The paper explained**

**Problem**
Pathogenic variants of FBXL4 cause an encephalopathic syndrome in children accompanied by lactic acidosis and mitochondrial DNA depletion (MTDPS13). The pathophysiology of the disease and the function of Fbxl4 are poorly understood.

**Results**
Here, we combined three different models, i.e., knockout mice, patient-derived fibroblasts, and a CRISPR/Cas9 knockout human cell line, to study the physiological role of Fbxl4. We report that Fbxl4 is involved in mitochondrial quality control and that its absence causes an increased lysosomal turnover of mitochondria leading to a decreased cellular mitochondrial content. Although the remaining mitochondria are fully functional, their numbers are insufficient to prevent disease.

**Impact**
The obtained results shed light on the pathophysiology of the disease and suggest that Fbxl4 participates in mitochondrial quality control. Interventions to stop the increased mitochondrial turnover should be considered as a potential treatment for this disease.

p62-directed metabolic reprogramming is essential for normal neurodifferentiation. *Stem Cell Rep* 12: 696–711

Chen Z, Liu L, Cheng Q, Li Y, Wu H, Zhang W, Wang Y, Sehgal SA, Siraj S, Wang X *et al* (2017) Mitochondrial E3 ligase MARCH5 regulates FUNDC1 to fine-tune hypoxic mitophagy. *EMBO Rep* 18: 495–509

Chomyn A (1996) *In vivo* labeling and analysis of human mitochondrial translation products. *Methods Enzymol* 264: 197–211

Cox J, Mann M (2008) MaxQuant enables high peptide identification rates, individualized p.p.b.-range mass accuracies and proteome-wide protein quantification. *Nat Biotechnol* 26: 1367–1372

Cox J, Neuhauser N, Michalski A, Scheltema RA, Olsen JV, Mann M (2011) Andromeda: a peptide search engine integrated into the MaxQuant environment. *J Proteome Res* 10: 1794–1805

Cox J, Hein MY, Luber CA, Paron I, Nagaraj N, Mann M (2014) Accurate proteome-wide label-free quantification by delayed normalization and maximal peptide ratio extraction, Termed MaxLFQ. *Mol Cell Proteomics* 13: 2513–2526

Dai H, Zhang VW, El-Hattab AW, Ficicioglu C, Shinawi M, Lines M, Schulze A, McNutt M, Gotway G, Tian X *et al* (2017) FBXL4 defects are common in patients with congenital lactic acidemia and encephalomyopathic mitochondrial DNA depletion syndrome. *Clin Genet* 91: 634–639

Di Rita A, Peschiaroli A, D'Acunzo P, Strobbe D, Hu Z, Gruber J, Nygaard M, Lambrughi M, Melino G, Papaleo E *et al* (2018) HUWE1 E3 ligase promotes PINK1/PARKIN-independent mitophagy by regulating AMBRA1 activation via IKKα. *Nat Commun* 9: 3755

El-Hattab AW, Dai H, Almannai M, Wang J, Faqeih EA, Al Asmari A, Saleh MAM, Elamin MAO, Alfadhel M, Alkuraya FS *et al* (2017) Molecular and clinical spectra of FBXL4 deficiency. *Hum Mutat* 38: 1649–1659

Filograna R, Koolmeister C, Upadhyay M, Pajak A, Clemente P, Wibom R, Simard ML, Wredenberg A, Freyer C, Stewart JB *et al* (2019) Modulation of mtDNA copy number ameliorates the pathological consequences of a heteroplasmic mtDNA mutation in the mouse. *Sci Adv* 5: eaav9824

Gai X, Ghezzi D, Johnson MA, Biagosch CA, Shamseldin HE, Haack TB, Reyes A, Tsukikawa M, Sheldon CA, Srinivasan S *et al* (2013) Mutations in FBXL4, encoding a mitochondrial protein, cause early-onset mitochondrial encephalomyopathy. *Am J Hum Genet* 93: 482–495

Gorman GS, Chinnery PF, DiMauro S, Hirano M, Koga Y, McFarland R, Suomalainen A, Thorburn DR, Zeviani M, Turnbull DM (2016) Mitochondrial diseases. *Nat Rev Dis Primers* 2: 16080

Hauser DN, Hastings TG (2013) Mitochondrial dysfunction and oxidative stress in Parkinson's disease and monogenic parkinsonism. *Neurobiol Dis* 51: 35–42

Hernandez DG, Reed X, Singleton AB (2016) Genetics in Parkinson disease: Mendelian versus non-Mendelian inheritance. *J Neurochem* 139(Suppl): 59–74

Huber W, von Heydebreck A, Sültmann H, Poustka A, Vingron M (2002) Variance stabilization applied to microarray data calibration and to the quantification of differential expression. *Bioinformatics* 18(Suppl 1): S96–S104

Huemer M, Karall D, Schossig A, Abdenur JE, Al Jasmi F, Biagosch C, Distelmaier F, Freisinger P, Graham BH, Haack TB *et al* (2015) Clinical, morphological, biochemical, imaging and outcome parameters in 21 individuals with mitochondrial maintenance defect related to FBXL4 mutations. *J Inherit Metab Dis* 38: 905–914

Kühl I, Miranda M, Atanassov I, Kuznetsova I, Hinze Y, Mourier A, Filipovska A, Larsson NG (2017) Transcriptomic and proteomic landscape of mitochondrial dysfunction reveals secondary coenzyme Q deficiency in mammals. *Elife* 6: e30952

Kuma A, Hatano M, Matsui M, Yamamoto A, Nakaya H, Yoshimori T, Ohsumi Y, Tokuhisa T, Mizushima N (2004) The role of autophagy during the early neonatal starvation period. *Nature* 432: 1032–1036

Kuma A, Komatsu M, Mizushima N (2017) Autophagy-monitoring and autophagy-deficient mice. *Autophagy* 13: 1619–1628

Larsson N-G, Wang J, Wilhelmsson H, Oldfors A, Rustin P, Lewandoski M, Barsh GS, Clayton DA (1998) Mitochondrial transcription factor A is necessary for mtDNA maintenance and embryogenesis in mice. *Nat Genet* 18: 231–236

Lavie J, De Belvalet H, Sonon S, Ion AM, Dumon E, Melser S, Lacombe D, Dupuy J-W, Lalou C, Bénard G (2018) Ubiquitin-dependent degradation of mitochondrial proteins regulates energy metabolism. *Cell Rep* 23: 2852–2863

Lee JJ, Sanchez-Martinez A, Zarate AM, Benincá C, Mayor U, Clague MJ, Whitworth AJ (2018) Basal mitophagy is widespread in *Drosophila* but minimally affected by loss of Pink1 or parkin. *J Cell Biol* 217: 1613–1622

Li Q, Li Y, Wang X, Qi J, Jin X, Tong H, Zhou Z, Zhang ZC, Han J (2017) Fbxl4 serves as a clock output molecule that regulates sleep through promotion of rhythmic degradation of the GABAA receptor. *Curr Biol* 27: 3616–3625.e5

Matic S, Jiang M, Nicholls TJ, Uhler JP, Dirksen-Schwanenland C, Polosa PL, Simard M-L, Li X, Atanassov I, Rackham O *et al* (2018) Mice lacking the mitochondrial exonuclease MGME1 accumulate mtDNA deletions without developing progeria. *Nat Commun* 9: 1202

McWilliams TG, Prescott AR, Allen GFG, Tamjar J, Munson MJ, Thomson C, Muqit MMK, Ganley IG (2016) mito-QC illuminates mitophagy and mitochondrial architecture *in vivo*. *J Cell Biol* 214: 333–345

Mizushima N, Levine B (2010) Autophagy in mammalian development and differentiation. *Nat Cell Biol* 12: 823–830

Montava-Garriga L, Singh F, Ball G, Ganley IG (2020) Semi-automated quantitation of mitophagy in cells and tissues. *Mech Ageing Dev* 185: 111196

Moon HE, Paek SH (2015) Mitochondrial dysfunction in Parkinson's disease. *Exp Neurobiol* 24: 103–116

Nelson DE, Randle SJ, Laman H (2013) Beyond ubiquitination: the atypical functions of Fbxo7 and other F-box proteins. *Open Biol* 3: 130131

Perez-Riverol Y, Csordas A, Bai J, Bernal-Llinares M, Hewapathirana S, Kundu DJ, Inuganti A, Griss J, Mayer G, Eisenacher M *et al* (2019) The PRIDE database and related tools and resources in 2019: improving support for quantification data. *Nucleic Acids Res* 47: D442–D450

Pickles S, Vigié P, Youle RJ (2018) Mitophagy and quality control mechanisms in mitochondrial maintenance. *Curr Biol* 28: R170–R185

R Development Core Team (2010) *R: a language and environment for statistical computing*. Vienna, Austria: R foundation for Statistical Computing

Ran FA, Hsu PD, Wright J, Agarwala V, Scott DA, Zhang F (2013) Genome engineering using the CRISPR-Cas9 system. *Nat Protoc* 8: 2281–2308

Rappsilber J, Ishihama Y, Mann M (2003) Stop and go extraction tips for matrix-assisted laser desorption/ionization, nanoelectrospray, and LC/MS sample pretreatment in proteomics. *Anal Chem* 75: 663–670

Read R, Savelieva K, Baker K, Hansen G, Vogel P (2011) Histopathological and neurological features of Atg4b knockout mice. *Vet Pathol* 48: 486–494

Ritchie ME, Phipson B, Wu D, Hu Y, Law CW, Shi W, Smyth GK (2015) limma powers differential expression analyses for RNA-sequencing and microarray studies. *Nucleic Acids Res* 43: e47

Skaar JR, Pagan JK, Pagano M (2013) Mechanisms and function of substrate recruitment by F-box proteins. *Nat Rev Mol Cell Biol* 14: 369–381

Sliter DA, Martinez J, Hao L, Chen X, Sun N, Fischer TD, Burman JL, Li Y, Zhang Z, Narendra DP *et al* (2018) Parkin and PINK1 mitigate STING-induced inflammation. *Nature* 561: 258–262

Sugiura A, McLelland G-L, Fon EA, McBride HM (2014) A new pathway for mitochondrial quality control: mitochondrial-derived vesicles. *EMBO J* 33: 2142 – 2156

Van Rechem C, Black JC, Abbas T, Allen A, Rinehart CA, Yuan G-C, Dutta A, Whetstine JR (2011) The SKP1-cul1-F-box and leucine-rich repeat protein 4 (SCF-FbxL4) ubiquitin ligase regulates lysine demethylase 4A (KDM4A)/Jumonji domain-containing 2A (JMJD2A) protein. *J Biol Chem* 286: 30462 – 30470

Wibom R, Hagenfeldt L, von Döbeln U (2002) Measurement of ATP production and respiratory chain enzyme activities in mitochondria isolated from small muscle biopsy samples. *Anal Biochem* 311: 139 – 151

Wortmann SB, Koolen DA, Smeitink JA, van den Heuvel L, Rodenburg RJ (2015) Whole exome sequencing of suspected mitochondrial patients in clinical practice. *J Inherit Metab Dis* 38: 437 – 443

Zheng N, Zhou Q, Wang Z, Wei W (2016) Recent advances in SCF ubiquitin ligase complex: clinical implications. *Biochim Biophys Acta* 1866: 12 – 22

Zhou ZD, Lee JCT, Tan EK (2018) Pathophysiological mechanisms linking F-box only protein 7 (FBXO7) and Parkinson's disease (PD). *Mutat Res* 778: 72 – 78

Zimmermann M, Reichert AS (2017) How to get rid of mitochondria: crosstalk and regulation of multiple mitophagy pathways. *Biol Chem* 399: 29 – 45

