## [Review Process File · EMBO Molecular Medicine]

FBXL4 deficiency increases mitochondrial removal by autophagy

David Alsina, Oleksandr Lytovchenko, Aleksandra Schab, Ilian Atanassov, Florian Schober, Min Jiang, Camilla Koolmeister, Anna Wedell, Robert Taylor, Anna Wredenberg, and Nils-Göran Larsson

DOI: [10.15252/emmm.201911659](https://doi.org/10.15252/emmm.201911659)

Corresponding author(s): Nils-Göran Larsson (Nils-Goran.Larsson@ki.se)

Review Timeline:

Submission Date:	24th Oct 19
Editorial Decision:	5th Dec 19
Revision Received:	8th Apr 20
Editorial Decision:	30th Apr 20
Revision Received:	11th May 20
Accepted:	14th May 20

Editor: Celine Carret

Transaction Report:

5th Dec 2019

Dear Prof. Larsson,

Thank you for the submission of your manuscript to EMBO Molecular Medicine and for your patience. We have now finally heard back from the three referees whom we asked to evaluate your manuscript.

You will see that the three referees find the study to be of interest. However, while ref. #3 is enthusiastic and recommends acceptance as is, refs. #1 and #2 are more critical and would like to see some mechanism added to the study. Both referees 1 and 2 provide interesting suggestions that if followed would support and strengthen the data. We therefore would like to encourage you to add some of these experiments and answer/discuss the questions asked in a revised article. Of particular relevance, we would encourage you to show the increased autophagic flux, independent of LC3 using independent techniques. Further, as the role of FBXL4 in autophagy/mitophagy is unclear, further evidence to support that mitochondrial dysfunction triggers increased autophagy, which, in turn reduces mitochondrial content should be provided.

We would therefore welcome the submission of a revised version within three months for further consideration and would like to encourage you to address all the criticisms raised as suggested to improve conclusiveness and clarity. Please note that EMBO Molecular Medicine strongly supports a single round of revision and that, as acceptance or rejection of the manuscript will depend on another round of review, your responses should be as complete as possible.

I look forward to receiving your revised manuscript.

Yours sincerely,

Celine Carret

Celine Carret, PhD
Senior Editor
EMBO Molecular Medicine

*** Instructions to submit your revised manuscript ***

**** PLEASE NOTE **** As part of the EMBO Publications transparent editorial process initiative (see our Editorial at <https://www.embopress.org/doi/pdf/10.1002/emmm.201000094>), EMBO Molecular Medicine will publish online a Review Process File to accompany accepted manuscripts.

To submit your manuscript, please follow this link:

Link Not Available

- 1) a .doc formatted version of the manuscript text (including Figure legends and tables). Please make sure that the changes are highlighted to be clearly visible to referees and editors alike.
- 2) separate figure files*
- 3) supplemental information as Expanded View and/or Appendix. Please carefully check the authors guidelines for formatting Expanded view and Appendix figures and tables at <https://www.embopress.org/page/journal/17574684/authorguide#expandedview>
- 4) a letter INCLUDING the reviewers' reports and your detailed responses to their comments (as Word file)

Also, and to save some time should your paper be accepted, please read below for additional information regarding some features of our research articles:

- 5) The paper explained: EMBO Molecular Medicine articles are accompanied by a summary of the articles to emphasize the major findings in the paper and their medical implications for the non-specialist reader. Please provide a draft summary of your article highlighting
 - the medical issue you are addressing,
 - the results obtained and
 - their clinical impact.

6) For more information: There is space at the end of each article to list relevant web links for further consultation by our readers. Could you identify some relevant ones and provide such information as well? Some examples are patient associations, relevant databases, OMIM/proteins/genes links, author's websites, etc...

7) Author contributions: the contribution of every author must be detailed in a separate section (before the acknowledgments).

8) EMBO Molecular Medicine now requires a complete author checklist (<https://www.embopress.org/page/journal/17574684/authorguide>) to be submitted with all revised manuscripts. Please use the checklist as a guideline for the sort of information we need WITHIN the manuscript as well as in the checklist. This is particularly important for animal reporting, antibody dilutions (missing) and exact p-values and n that should be indicated instead of a range.

9) Every published paper now includes a 'Synopsis' to further enhance discoverability. Synopses are displayed on the journal webpage and are freely accessible to all readers. They include a short stand first (maximum of 300 characters, including space) as well as 2-5 one sentence bullet points that summarise the paper. Please write the bullet points to summarise the key NEW findings. They should be designed to be complementary to the abstract - i.e. not repeat the same text. We encourage inclusion of key acronyms and quantitative information (maximum of 30 words / bullet point). Please use the passive voice. Please attach these in a separate file or send them by email, we will incorporate them accordingly.

You are also welcome to suggest a striking image or visual abstract to illustrate your article. If you do please provide a jpeg file 550 px-wide x 400-px high.

10) A Conflict of Interest statement should be provided in the main text

11) Please note that we now mandate that all corresponding authors list an ORCID digital identifier. This takes <90 seconds to complete. We encourage all authors to supply an ORCID identifier, which will be linked to their name for unambiguous name identification.

Currently, our records indicate that the ORCID for your account is 0000-0001-5100-996X.

Link Not Available

12) The system will prompt you to fill in your funding and payment information. This will allow Wiley to send you a quote for the article processing charge (APC) in case of acceptance. This quote takes into account any reduction or fee waivers that you may be eligible for. Authors do not need to pay any fees before their manuscript is accepted and transferred to our publisher.

Photos 400-800 DPI

*Additional important information regarding figures and illustrations can be found at <http://bit.ly/EMBOPressFigurePreparationGuideline>

***** Reviewer's comments *****

Referee #1 (Comments on Novelty/Model System for Author):

The experiments are elegant and the results convincing. I am less convinced of the medical impact because the real meaning of increased autophagic degradation is unclear, and the idea of interfering with autophagy to restore mitochondrial content sounds risky. The cellular and animal models used are appropriate.

Referee #1 (Remarks for Author):

Alsina and colleagues report here an unexpected role for FBXL4 in the quality control of mitochondria. Mutations in FBXL4 have been found in a relatively high number of patients affected by mtDNA depletion syndrome and encephalomyopathy. However, the role of FBXL4 in mtDNA maintenance remains unclear.

FBXL4 belongs to the family of F-box proteins which normally serve as substrate adaptors for the Skp-Cullin-F-box E3 ubiquitin ligases. However, there is no evidence that FBXL4 interacts with SCF proteins and that it is associated with ubiquitination.

In order to investigate the role of Fbxl4 in mtDNA maintenance, the authors initially generated and characterized a Fbxl4 KO mouse. A high level of embryonic lethality was observed, but, surprisingly, the surviving KO mice were normal up to 8-12 months, when they started to lose weight and developed a prominent hunchback. MtDNA was decreased in several organs, as well as mtDNA-encoded transcripts, while nDNA-encoded mitochondrial transcripts were normal. Both mtDNA- and nDNA-encoded mitochondrial proteins were reduced by western blot and quantitative proteomics, while the latter approach revealed also an increase of lysosomal proteins. These findings are compatible with an increased turnover of mitochondria by autophagy.

To further investigate this aspect, the authors turned to cellular models, including fibroblasts from a newly identified patient, carrying new mutations in FBXL4, and knockout cells generated by CRISPR/Cas9 technology. Both cell lines showed features compatible with the phenotype described in mice, including reduced mtDNA and mtDNA-encoded proteins content, and increased levels of lysosomal markers.

In cell translation experiments in FBXL4 KO cells showed that mitochondrial protein synthesis was normal, while degradation was increased in the absence of Fbxl4. This difference was more evident in the presence of the proteasomal inhibitor epoxomicin, while ammonium chloride rescued the phenotype. This suggests that FBXL4 is involved in the lysosomal degradation of mitochondria. How do the authors explain that epoxomicin "increased the differences in protein stability between wild-type and knockout cells"? Is it because the degradation through autophagy is increased? Finally, the authors analysed the autophagic flux in the presence/absence of ammonium chloride. p62 and LC3-II accumulated to the same levels in the presence of lysosomal blockers.

The paper is interesting, the experiment well conducted, and the interpretation fits with the data. However, I have some concerns on the real meaning of these findings, which seem to be rather observational to this reviewer, because the role of Fbxl4 in autophagy is not explained or explored. In addition, there is an accumulating body of evidence suggesting that mitochondrial dysfunction may impact on lysosomal activity, leading eventually to reduced autophagic flux. It may well be that in FBXL4 mutants the effect on autophagy is the opposite, but I think more solid evidence for this should be provided.

I have some specific comments and suggestions that the authors should consider.

First, FBXL4 patients, including the new one described here, are characterized by encephalomyopathy with reduced OXPHOS activities in skeletal muscle. Is mtDNA reduced in the skeletal muscle of the patient and of the mouse? Does this result in an OXPHOS defect in the mouse?

Second, the authors report some neurological/behavioural abnormalities in FBXL4: did the authors observed any neuropathological alteration in the knockout brains?

Third, the reduction of respiratory complexes subunits does not seem to lead to an OXPHOS dysfunction, but this has only been tested by BNAGE in gel activity. A more quantitative method (spectrophotometry or oxygen consumption) should be used.

Fourth, the analysis of lysosomal degradation of mitochondria is based exclusively on proteomic quantification, and does not seem to be confirmed by western blot (Figure 3F). A more detailed analysis of lysosomal function should be used. Fluorescent probes to analyse lysosomal pH, such as Oregon Green, are commercially available, and several methods are described in the literature (see for instance Fernandez-Mosquera et al, Autophagy, 2018).

Fifth, a quantification of the bands for LC3-II, and eventually p62, should be included in figure S3. Although the authors say that LC3-II levels are similar in wild-type and knockout cells, I am not completely sure that LC3-II levels are not decreased in the knockout cells in the absence of FBS. Densitometric quantification would be helpful to correctly interpret the results. In addition, the analysis of the autophagic flux by an alternative method, for instance by using suitable probes such mtKeima or MitoQC, would confirm by an independent method the results obtained by using ammonium chloride.

Referee #2 (Comments on Novelty/Model System for Author):

Sufficient number of state-of-the-art methods and approaches used; Important biological mechanism explained; translation into the clinic as difficult as always.

Referee #2 (Remarks for Author):

The most common cause of congenital lactic acidosis in children are defects in the pyruvate dehydrogenase complex. Previous studies using patient-derived fibroblasts have shown that mutations in FBXL4 cause decreased levels of OXPHOS proteins, low oxygen consumption etc., but the pathomechanism has been unknown until now. Mutations in FBXL4 make up 14 % of such cases and therefore, mutations in this protein are one of the most common causes of mitochondrial disease.

In this paper, an FBXL4 knockout mouse has been generated and the mitochondrial phenotype has been compared to patient-derived fibroblasts with mutations in FBXL4 as well to knockout cells. In summary, the paper provides strong evidence that FBXL4 mutations lead to an increased lysosomal activity followed by depletion of mitochondria due to enhanced autophagy, thereby finally mimicking a mitochondrial disease.

The paper is very well written and the data are clearly in strong favor of this hypothesis.

Major points of criticism:

What is unclear to me is how the enrichment of lysosomal proteins is explained. Is FBXL4 a protein involved in lysosomal biogenesis, or does it also influence lysosomal turnover? For example, in the sentence "The increased levels of lysosomal proteins ... argues ... that increased lysosomal degradation ... may explain ..." (page 8, line 193), the causality is completely unclear.

Also, there is no increase in lysosomal proteins like LAMP1, LAMP2, etc. shown by WB or proteomics. This would mean that there must be an increase in lysosomal activity more than in lysosomal content. Immunofluorescence for Lamp1/2 to visualize lysosomal structure or the use of LysoSensor for lysosomal pH could be helpful.

The paper could be strengthened considerably by adding more approaches to study mitochondrial turnover, which is now based on one simple pulse-chase experiment (Fig. 4c). There are techniques to test this, i.e. mitoTIMER, mitochondrial-GFP-RFP, colocalization mitochondria-LC3 etc. This would be especially important since in this case mitochondrial turnover is obviously due to an alternative pathway which is independent of LC3 conversion.

If general autophagy is normal (no increase in flux), but mitochondrial turnover is increased, I would expect that Ubiquitin, p62 or LC3 are more abundant on mitochondria. I recommend mitochondria isolation in order to check for this.

The lysosomal blocker NH₄Cl neutralizes lysosomal pH, therefore only the latest step of autophagy. The authors claim that turnover is independent of autophagy because there is no increase in autophagic flux. If this is true, the authors should check for example with 3MA, which blocks early steps of autophagy (formation of autophagosome).

If the increased turnover is not due to classical autophagy, why would inhibition of the classic pathway be beneficial and even a possible treatment for patients? This should be explained in more detail.

Minor points:

The statistical methods are not explained in the methods section and also not in the Figure legends, just p values are given.

1. Figure 1b: no statistics shown, here a chi² test may be appropriate

2. Figure 2b: Was a t-test WT Vs mut performed for each gene and for each tissue? I would rather do an ANOVA, either one way to compare inside tissue or two way if they want to compare also tissues.

3. Figure 3b: No statistics; here it should be a t-test; also no statistics in 3d or 3e

Referee #3 (Comments on Novelty/Model System for Author):

NG Larsson laboratory has been a leader in developing mouse models relevant to human genetic disease. In this MS they characterize a novel FBXL4 mouse model of the human disease and suggest a pathophysiological mechanism, with laboratory support, of the mechanism by which mitochondrial biogenesis is decreased.

The MS is technically sound, and introduces new data to suggest mechanism by which FBXL4 affects both mitochondrial biogenesis and also neurological issues in humans.

Referee #3 (Remarks for Author):

There appear to be some minor grammatical/syntactical/word choice errors, i.e. it should be a 'brake' rather than a 'break' on mitochondrial autophagy.

Dear Editor,

we would like to thank the three referees for providing expert input on our manuscript. We also appreciate the positive comments: “*The experiments are elegant and the results convincing – Referee 1*”, “*The paper is very well written and the data are clearly in strong favor of this hypothesis – Referee 2*”. We have addressed the comments of the referees in point by point response below. Some figures have been updated and changes in the manuscript text are highlighted in red.

Referee #1

How do the authors explain that epoxomicin "increased the differences in protein stability between wild-type and knockout cells"? Is it because the degradation through autophagy is increased?

Indeed, inhibition of the proteasome by epoxomicin activates autophagy as is shown by the increased levels of LC3-II and decreased levels of p62. This effect has also been shown by other groups (for example in Demishtein et al. 2017 Autophagy).

I have some specific comments and suggestions that the authors should consider.

First, FBXL4 patients, including the new one described here, are characterized by encephalomyopathy with reduced OXPHOS activities in skeletal muscle. Is mtDNA reduced in the skeletal muscle of the patient and of the mouse?

Most of the patients show decreased mtDNA levels in muscle, however the patient identified at Karolinska hospital, named as Patient 3 in this study, did not show any significant decrease in mtDNA copy number in this tissue. In the mouse, we analyzed the mtDNA levels in skeletal muscle at 1 year of age and did not observe any significant changes. Figure 2A has been updated to include these data.

Does this result in an OXPHOS defect in the mouse?

We performed sequential COX/SDH histochemistry in skeletal muscle sections to address this question. In agreement with the mtDNA copy number not being affected in this tissue, we did not observe any blue (COX-deficient, SDH-positive) fibers following the COX/SDH reaction or any other clear difference between wild-type and KO animals. We also subjected sections to the individual COX enzyme reaction but did not observe any significant differences. These data have been included in Supplementary Figure 1. The Methods section has been updated as well.

Second, the authors report some neurological/behavioural abnormalities in FBXL4: did the authors observe any neuropathological alteration in the knockout brains?

This is a really interesting point since most of the patients show abnormalities in the brain. We analyzed brain sections from 3 knock-out animals and 3 wild-type animals and looked for malformations and/or abnormalities. However, despite some defects in one of the animals, slight enlargement of lateral ventricles, we did not observe any common pattern. It is thus clear from our experiments that there is no general occurrence of brain malformations/abnormalities in *Fbxl4* knockout mice. However, the low number of analyzed

animals does not allow us to make any solid statement about a possible increase in the frequency of brain malformations/abnormalities in the knockout mice. We will address this interesting aspect in a future study where we will breed a large cohort of *Fbxl4* knockout mice for extensive neurological phenotyping combined with histology and molecular analyses of brain. Given the low number of knockout animals that comes through the germ line this experiment will require 3-4 years to complete.

From the molecular point of view, we present new data showing a clear decrease in mtDNA copy number (Figure 2A) and mitochondrial transcript levels (Figure 2B), which could potentially impair neuronal function. A separate study as outlined above will be necessary to address this question.

Third, the reduction of respiratory complexes subunits does not seem to lead to an OXPHOS dysfunction, but this has only been tested by BNGE in gel activity. A more quantitative method (spectrophotometry or oxygen consumption) should be used.

To address this question, we performed spectrophotometric analyses of respiratory chain enzyme activities in isolated mouse liver mitochondria. As shown in the updated Figure S2, we did not observe any difference in respiratory chain complex activities as normalized to citrate synthase, with the exception of a rather mild (86% residual activity) but significant decrease of complex IV activity in *Fbxl4* knockout liver mitochondria. The results and methods sections have been updated with these new results.

Fourth, the analysis of lysosomal degradation of mitochondria is based exclusively on proteomic quantification, and does not seem to be confirmed by western blot (Figure 3F). A more detailed analysis of lysosomal function should be used. Fluorescent probes to analyse lysosomal pH, such as Oregon Green, are commercially available, and several methods are described in the literature (see for instance Fernandez-Mosquera et al, Autophagy, 2018). We performed LysoSensor Green staining in the different fibroblast lines, and we did not observe significant changes, suggesting that the lysosomal activity is the same. However, we observed increased levels of several lysosomal proteins in mouse liver and patient fibroblast (Fig. 2E-F and Fig. 3C-D). Based on these results, we conclude that increased lysosomal mass is the driving force in the observed mitochondrial turnover, whereas lysosomal pH is not affected. The LysoSensor results have been added as a supplementary data (Fig. S3).

Fifth, a quantification of the bands for LC3-II, and eventually p62, should be included in figure S3. Although the authors say that LC3-II levels are similar in wild-type and knockout cells, I am not completely sure that LC3-II levels are not decreased in the knockout cells in the absence of FBS. Densitometric quantification would be helpful to correctly interpret the results.

LC3-II and p62 bands have been quantified and the data is included in Figure S4. No significant difference was observed between control and patient fibroblast lines.

In addition, the analysis of the autophagic flux by an alternative method, for instance by using suitable probes such mtKeima or MitoQC, would confirm by an independent method the results obtained by using ammonium chloride

We expressed the mitoQC probe in the different patient fibroblast lines and in a control line and used confocal microscopy to investigate the presence of mitolysosomes (only-red signal). We observed a slight, but significant, increase of mitolysosomes/cell area in the patient lines compared to the control line. This result is in good agreement with other results shown in the manuscript and are described in a new section “*Fibroblasts from FBXL4-deficient patients show increased mitophagy*” – page 10 line 274, and in the revised Figure 5.

Referee #2

Major points:

What is unclear to me is how the enrichment of lysosomal proteins is explained. Is FBXL4 a protein involved in lysosomal biogenesis, or does it also influence lysosomal turnover? For example, in the sentence "The increased levels of lysosomal proteins ... argues ... that increased lysosomal degradation ... may explain ..." (page 8, line 193), the causality is completely unclear.

We apologize for not making this clearer in the manuscript. We have now clarified this issue (page 8, line 198 and page 10, line 226).

Also, there is no increase in lysosomal proteins like LAMP1, LAMP2, etc. shown by WB or proteomics. This would mean that there must be an increase in lysosomal activity more than in lysosomal content. Immunofluorescence for Lamp1/2 to visualize lysosomal structure or the use of Lysosensor for lysosomal pH could be helpful.

On westerns, we found a slight non-significant ($p < 0.09$) increase in Lamp1 protein levels in patient fibroblasts) in comparison with control fibroblasts (Figure 3C-D). However, the data from proteomics clearly show increased levels of several lysosomal proteins in mouse liver and patient fibroblast lacking FBXL4 (Fig. 2E-F and Fig. 3C-D), which indeed indicates that the lysosomal protein content is increased. We added a supplementary figure with Lamp2 immunofluorescence and Lysosensor Green fluorescence measurements in the different fibroblast lines (Fig S3). We could not find any significant difference in lysosomal morphology or pH between control and patient lines. In summary, our results argue that the lysosomal content is higher in cells lacking FBXL4.

The paper could be strengthened considerably by adding more approaches to study mitochondrial turnover, which is now based on one simple pulse-chase experiment (Fig. 4c). There are techniques to test this, i.e. mitoTIMER, mitochondrial-GFP-RFP, colocalization mitochondria-LC3 etc. This would be especially important since in this case mitochondrial turnover is obviously due to an alternative pathway which is independent of LC3 conversion. As mentioned above, we expressed mitoQC in the patient fibroblasts lines and confirmed an increased mitochondrial turnover when compared to a control line.

If general autophagy is normal (no increase in flux), but mitochondrial turnover is increased, I would expect that Ubiquitin, p62 or LC3 are more abundant on mitochondria. I recommend mitochondria isolation in order to check for this.

We performed this experiment using sucrose gradient purified mitochondria from the different fibroblast lines. The obtained results show no significant changes between controls and patients but there was high variability between the different lines (also between the controls). These results support the idea that the pathway involved in the increased mitochondrial turnover is independent of LC3 conversion, as observed in the autophagic flux experiment, thus pointing to other pathways like Rab-mediated alternative autophagy, MDVs or micromitophagy.

The lysosomal blocker NH₄Cl neutralizes lysosomal pH, therefore only the latest step of autophagy. The authors claim that turnover is independent of autophagy because there is no increase in autophagic flux. If this is true, the authors should check for example with 3MA, which blocks early steps of autophagy (formation of autophagosome).

The 3-methyl adenine (3MA) compound has been reported to have a dual role and can both activate and inhibit autophagy, see e.g. Wu et al. JBC (2010). Under our experimental conditions we found that 3MA treatment causes an increase of LC3-II. We are very interested in dissecting the pathway that leads to increased mitochondrial turnover in the absence of Fbxl4, but we believe that a rather comprehensive genetic approach is needed in the mouse to clarify this issue.

If the increased turnover is not due to classical autophagy, why would inhibition of the classic pathway be beneficial and even a possible treatment for patients? This should be explained in more detail.

In the last paragraph of the Discussion section we emphasize that this hypothetical treatment should be designed to reduce mitochondrial clearance by autophagy not to stop autophagy in general.

Minor points:

The statistical methods are not explained in the methods section and also not in the Figure legends, just p values are given.

We apologize for this, we added a statistics section in materials and methods.

1. Figure 1b: no statistics shown, here a chi² test may be appropriate

We have performed this analysis and it is now included in the results section.

2. Figure 2b: Was a t-test WT Vs mut performed for each gene and for each tissue?

I would rather do an ANOVA, either one way to compare inside tissue or two way if they want to compare also tissues.

As we compare each gene in each tissue we feel the t-test is appropriate.

3. Figure 3b: No statistics; here it should be a t-test; also no statistics in 3d or 3e

Figure has been updated with the statistics; a t-test has been performed.

Referee #3

There appear to be some minor grammatical/syntactical/word choice errors, i.e. it should be a 'brake' rather than a 'break' on mitochondrial autophagy

We went through the manuscript and corrected the mistakes.

30th Apr 2020

Dear Prof. Larsson,

Thank you for the submission of your revised manuscript to EMBO Molecular Medicine. We have now received the enclosed reports from the referees that were asked to re-assess it. As you will see the reviewers are now supportive and I am pleased to inform you that we will be able to accept your manuscript pending the following final amendments:

1) Tables

Please change the format and type of the table such as Tables EV1, EV2 and EV3 are uploaded as "dataset" and called Dataset EV1, EV2 and EV3.

Table EV4 should be added to the Appendix document and labeled as Appendix Table S4. Please modify call outs accordingly.

2) Figures

Please change the nomenclature of Figures S1 to S4 to Figures EV1 to EV4. Expanded View Figure Legends must be in the main article, after Figure Legends.

3) In the main manuscript file, please do the following:

- correct/answer the track changes suggested by our data editors by working from the uploaded document
- add up to 5 keywords
- in M&M, the statistical paragraph should reflect all information that you have filled in the Authors checklist, especially regarding randomisation, blinding, replication.
- indicate in legends exact n= and exact p= values, not a range, along with the statistical test used. Some people found that to keep the figures clear, providing an Appendix table Sx with all exact p-values was preferable. You are welcome to do this if you want to.
- in M&M, include a statement that informed consent was obtained from all human subjects from whom cells were used, and that the experiments conformed to the principles set out in the WMA Declaration of Helsinki and the Department of Health and Human Services Belmont Report.
- provide the origin of cell lines used
- in M&M, the PCR primers and probes used are missing and must be provided
- in M&M, provide the antibody dilutions that were used for each antibody
- in M&M, regarding the paragraph "Oxphos...", the reference 43 is not enough. Please provide more details
- References are not numbered but must be alphabetical. In the reference list have 10 names followed by et al.

4) Source Data:

Please merge together all the Source Data files related to the same figure, in this case figure S4.

5) For more information: There is space at the end of each article to list relevant web links for further consultation by our readers. Could you identify some relevant ones and provide such information as well? Some examples are patient associations, relevant databases, OMIM/proteins/genes links, author's websites, etc...

6) The Paper Explained: EMBO Molecular Medicine articles are accompanied by a summary of the articles to emphasize the major findings in the paper and their medical implications for the

non-specialist reader. Please provide a draft summary of your article highlighting:

- the medical issue you are addressing, = Problem
- the results obtained = Results
- their clinical impact = Impact

7) Every published paper now includes a 'Synopsis' to further enhance discoverability. Synopses are displayed on the journal webpage and are freely accessible to all readers. They include a short stand first (maximum of 300 characters, including space) as well as 2-5 one sentence bullet points that summarise the paper. Please write the bullet points to summarise the key NEW findings. They should be designed to be complementary to the abstract - i.e. not repeat the same text. We encourage inclusion of key acronyms and quantitative information (maximum of 30 words / bullet point). Please use the passive voice. Please attach these in a separate file or send them by email, we will incorporate them accordingly.

You are also encouraged to suggest a striking image or visual abstract to illustrate your article. If you do please provide a jpeg file 550 px-wide x (250-400)-px high.

8) As part of the EMBO Publications transparent editorial process initiative (see our Editorial at <http://embomolmed.embopress.org/content/2/9/329>), EMBO Molecular Medicine will publish online a Review Process File (RPF) to accompany accepted manuscripts.

In the event of acceptance, this file will be published in conjunction with your paper and will include the anonymous referee reports, your point-by-point response and all pertinent correspondence relating to the manuscript. Let us know whether you agree with the publication of the RPF.

9) Data and software availability:

To list the primary data generated in your study, we would kindly ask you to include a formal "Data and software availability" section (after Materials & Methods) that follows the example below:

Please use the following format to report the accession number of your data

- [data type]: [full name of the resource] [accession number/identifier] ([doi or URL or identifiers.org/DATABASE:ACCESSION])

ex: Protein interaction AP-MS data: PRIDE PXD000xxx
(<http://www.ebi.ac.uk/pride/archive/projects/PXD000xxx>)

I look forward to reading a new revised version of your manuscript as soon as possible.

Yours sincerely,

Celine Carret

Celine Carret, PhD
Senior Editor
EMBO Molecular Medicine

*** Instructions to submit your revised manuscript ***

To submit your manuscript, please follow this link:

Link Not Available

- 1) a .doc formatted version of the manuscript text (including Figure legends and tables)
- 2) Separate figure files*
- 3) supplemental information as Expanded View and/or Appendix. Please carefully check the authors guidelines for formatting Expanded view and Appendix figures and tables at <https://www.embopress.org/page/journal/17574684/authorguide#expandedview>
- 4) a letter INCLUDING the reviewer's reports and your detailed responses to their comments (as Word file).
- 5) The paper explained: EMBO Molecular Medicine articles are accompanied by a summary of the articles to emphasize the major findings in the paper and their medical implications for the non-specialist reader. Please provide a draft summary of your article highlighting
 - the medical issue you are addressing,
 - the results obtained and
 - their clinical impact.This may be edited to ensure that readers understand the significance and context of the research. Please refer to any of our published articles for an example.

6) For more information: There is space at the end of each article to list relevant web links for further consultation by our readers. Could you identify some relevant ones and provide such information as well? Some examples are patient associations, relevant databases, OMIM/proteins/genes links, author's websites, etc...

7) Author contributions: the contribution of every author must be detailed in a separate section.

8) EMBO Molecular Medicine now requires a complete author checklist (<https://www.embopress.org/page/journal/17574684/authorguide>) to be submitted with all revised manuscripts. Please use the checklist as guideline for the sort of information we need WITHIN the manuscript. The checklist should only be filled with page numbers where the information can be found. This is particularly important for animal reporting, antibody dilutions (missing) and exact values and n that should be indicated instead of a range.

9) Every published paper now includes a 'Synopsis' to further enhance discoverability. Synopses are displayed on the journal webpage and are freely accessible to all readers. They include a short stand first (maximum of 300 characters, including space) as well as 2-5 one sentence bullet points that summarise the paper. Please write the bullet points to summarise the key NEW findings. They should be designed to be complementary to the abstract - i.e. not repeat the same text. We encourage inclusion of key acronyms and quantitative information (maximum of 30 words / bullet point). Please use the passive voice. Please attach these in a separate file or send them by email, we will incorporate them accordingly.

You are also welcome to suggest a striking image or visual abstract to illustrate your article. If you do please provide a jpeg file 550 px-wide x 400-px high.

10) A Conflict of Interest statement should be provided in the main text

11) Please note that we now mandate that all corresponding authors list an ORCID digital identifier. This takes <90 seconds to complete. We encourage all authors to supply an ORCID identifier, which will be linked to their name for unambiguous name identification.

Currently, our records indicate that the ORCID for your account is 0000-0001-5100-996X.

Link Not Available

12) The system will prompt you to fill in your funding and payment information. This will allow Wiley to send you a quote for the article processing charge (APC) in case of acceptance. This quote takes into account any reduction or fee waivers that you may be eligible for. Authors do not need to pay any fees before their manuscript is accepted and transferred to our publisher.

Photos 400-800 DPI

*Additional important information regarding figures and illustrations can be found at <http://bit.ly/EMBOPressFigurePreparationGuideline>

The system will prompt you to fill in your funding and payment information. This will allow Wiley to send you a quote for the article processing charge (APC) in case of acceptance. This quote takes into account any reduction or fee waivers that you may be eligible for. Authors do not need to pay any fees before their manuscript is accepted and transferred to our publisher.

***** Reviewer's comments *****

Referee #1 (Comments on Novelty/Model System for Author):

The study relies on very elegant and carefully performed experiments. The concept of Fbx14 as a regulator of mitophagy is new. I have some (minor) concerns on the medical impact. However, the experiments on the human fibroblasts confirm the observations made on the mouse model.

Referee #1 (Remarks for Author):

The authors addressed all my concerns and I have no further comments on the manuscript.

14th May 2020

Dear Prof. Larsson,

Thank you for your fast response. We are pleased to inform you that your manuscript is accepted for publication and is now being sent to our publisher to be included in the next available issue of EMBO Molecular Medicine.

We would like to remind you that as part of the EMBO Publications transparent editorial process initiative, EMBO Molecular Medicine will publish a Review Process File online to accompany accepted manuscripts. If you do NOT want the file to be published or would like to exclude figures, please immediately inform the editorial office via e-mail.

Please be reminded that the dataset deposited in PRIDE must be made immediately available upon publication.

Please read below for additional IMPORTANT information regarding your article, its publication and the production process.

Congratulations on your interesting work,

Celine Carret

Celine Carret, PhD
Senior Editor
EMBO Molecular Medicine

Follow us on Twitter @EmboMolMed
Sign up for eTOCs at embopress.org/alertsfeeds

*** ** IMPORTANT INFORMATION ** **

SPEED OF PUBLICATION

The journal aims for rapid publication of papers, using using the advance online publication "Early View" to expedite the process: A properly copy-edited and formatted version will be published as "Early View" after the proofs have been corrected. Please help the Editors and publisher avoid delays by providing e-mail address(es), telephone and fax numbers at which author(s) can be contacted.

Should you be planning a Press Release on your article, please get in contact with embomolmed@wiley.com as early as possible, in order to coordinate publication and release dates.

LICENSE AND PAYMENT:

All articles published in EMBO Molecular Medicine are fully open access: immediately and freely available to read, download and share.

EMBO Molecular Medicine charges an article processing charge (APC) to cover the publication costs. You, as the corresponding author for this manuscript, should have already received a quote with the article processing fee separately. Please let us know in case this quote has not been received.

Once your article is at Wiley for editorial production you will receive an email from Wiley's Author Services system, which will ask you to log in and will present you with the publication license form for completion. Within the same system the publication fee can be paid by credit card, an invoice, pro forma invoice or purchase order can be requested.

Payment of the publication charge and the signed Open Access Agreement form must be received before the article can be published online.

PROOFS

You will receive the proofs by e-mail approximately 2 weeks after all relevant files have been sent to our Production Office. Please return them within 48 hours and if there should be any problems, please contact the production office at embopressproduction@wiley.com.

Please inform us if there is likely to be any difficulty in reaching you at the above address at that time. Failure to meet our deadlines may result in a delay of publication.

All further communications concerning your paper proofs should quote reference number EMM-2019-11659-V3 and be directed to the production office at embopressproduction@wiley.com.

Thank you,

Celine Carret, PhD
Senior Editor
EMBO Molecular Medicine

Corresponding Author Name: Nils-Göran Larsson

Manuscript Number: EMM-2019-11659